microbiology/bioinformatics

plague, source, history, *Yersinia pestis*, lexicometry, second pandemic

**Author for correspondence:**
Michel Drancourt
e-mail: michel.drancourt@univ-amu.fr

†These authors contributed equally to this study.

# Differential word expression analyses highlight plague dynamics during the second pandemic

Rémi Barbieri[1,2,3,†], Riccardo Nodari[4,†], Michel Signoli[2], Sara Epis[4], Didier Raoult[1,3] and Michel Drancourt[1,3]

[1]Aix Marseille Univ., IRD, MEPHI, IHU Méditerranée Infection, Marseille 13005, France
[2]UMR 7268, Anthropologie bioculturelle, Droit, Ethique et Santé, Aix Marseille Univ, 11 CNRS, EFS, ADES, Marseille 13344, France
[3]IHU Méditerranée Infection, Marseille 13005, France
[4]Department of Biosciences and Pediatric Clinical Research Center 'Romeo and Enrica Invernizzi', University of Milan, Milan 20133, Italy

MD, 0000-0003-0768-1139

Research on the second plague pandemic that swept over Europe from the fourteenth to nineteenth centuries mainly relies on the exegesis of contemporary texts and is prone to interpretive bias. By leveraging certain bioinformatic tools routinely used in biology, we developed a quantitative lexicography of 32 texts describing two major plague outbreaks, using contemporary plague-unrelated texts as negative controls. Nested, network and category analyses of a 207-word pan-lexicome, comprising overrepresented terms in plague-related texts, indicated that 'buboes' and 'carbuncles' are words that were significantly associated with the plague and signalled an ectoparasite-borne plague. Moreover, plague-related words were associated with the terms 'merchandise', 'movable', 'tatters', 'bed' and 'clothes'. Analysing ancient texts using the method reported in this paper can certify plague-related historical records and indicate the particularities of each plague outbreak, which can inform on the potential sources for the causative *Yersinia pestis*.

## 1. Introduction

Plague, a deadly zoonosis caused by the bacterium *Yersinia pestis* [1,2], has been incontrovertibly identified via palaeomicrobiological research on numerous historically described burial sites in Europe, ending decades-long controversies regarding the aetiology of the so-called 'Black Death' (1346–1353)

and the related episodes that formed the second plague pandemic, which lasted from 1346 until the nineteenth century [3–9]. The ancient texts related to these plague episodes reported massive mortality rates [10–13], with an estimated 30 million deaths attributed solely to the Black Death [6,13]. Similar figures have not been observed during the plague outbreaks from 1894 to the current third plague pandemic [14,15], leaving the forces that shaped the plague dynamics during the second plague pandemic somewhat speculative [15–18].

Readings of second plague pandemic descriptions have been made amidst the anachronistic background of cumulative observations from the third pandemic, which may not be applicable to the second pandemic [19,20]. Accordingly, an extensive analysis of the differences between the second and third plague pandemic historical descriptions demonstrated that the two pandemics exhibited such different epidemiological and clinical descriptions that they may even have to be considered two different diseases [21]. For example, the sources of plague during the second pandemic could be related to clothes and goods, with no reference to epizootic episodes among rats [21]. Additionally, the incidence of infection was higher for individuals living in the same household, suggesting that the plague was able to spread from humans to humans [22–24].

Therefore, the sources of and routes for the spread of *Y. pestis* in medieval European populations remain speculative, deriving as they do from the mathematical modelling of ancient plague episodes [15], which obviously cannot be verified.

In the next perspective of contributing to these matters, we here designed a novel quantitative method for the automatic analysis of ancient plague texts, derived from a method routinely used in biology to measure the relative expression of genes in a set of biological samples, including negative control samples [25]. This model uses ancient plague-related text sets and negative control texts to measure the differential expression of plague-related words, creating a core-lexicome (core signature) for the ancient plague and an accessory-lexicome that is specific to each ancient plague outbreak, thereby opening an avenue for studying the pan-lexicome of the plague. The method described here was developed based on positive control texts describing the palaeomicrobiologically confirmed Great Plague that affected the city of Marseille and Provence at large in 1720–1722 [4,8,26,27] as well as the plague episode that ravaged northern Italy in 1629–1631 [3,27–29].

# 2. Results

## 2.1. Building the Marseille word database

A word repertoire was drawn from 16 historical French texts describing the microbiologically documented 1720–1722 plague of Marseille, France [4,8,26] and from 11 contemporary French texts that are unrelated to the plague (electronic supplementary material, data files S1 and S2), following a verification that the maximum Jaccard index between any two texts was 0.08 using the 'Jaccard' package in Rstudio (electronic supplementary material, figure S1A). This ensured that the 27 texts were original enough (i.e. without significant plagiarism) to be used as independent samples. This repertoire comprised a total of 2 049 566 words directly imported (without any modification) from Google Books into Rstudio; 473 681 words (23.1%) were poorly digitized due to damage resulting from preservation conditions and therefore could not be included in further analysis. Finally, 1 575 885 words (76.9%), representative of 10 315 unique words, were cleaned by removing noncharacter letters and spaces, replacing ancient characters with homologous characters in the current alphabet (i.e. the 'long S' was replaced with 's'), filtering (with a French glossary of 336 631 words, conjugated verbs and proper nouns) and deleting words with fewer than five occurrences in all 27 texts (the complete script is available as electronic supplementary material, text S1). Altogether, the final 1 575 885-word database comprised 904 005 words from the 16 plague-related texts and 671 880 words from the 11 negative control texts.

## 2.2. Analysing the Marseille word database

The word database we created was used to compare the relative expression of words in the 16 plague-related texts and the 11 negative control texts (figure 2*a*). This comparison was performed by leveraging informatics methods and employing cut-off values commonly used in the field of transcriptomics (for the analysis of RNA sequences, i.e. the significant letter series using the universal biological code, in biological samples) (figure 1). Accordingly, using the DESeq2 package originally developed to identify

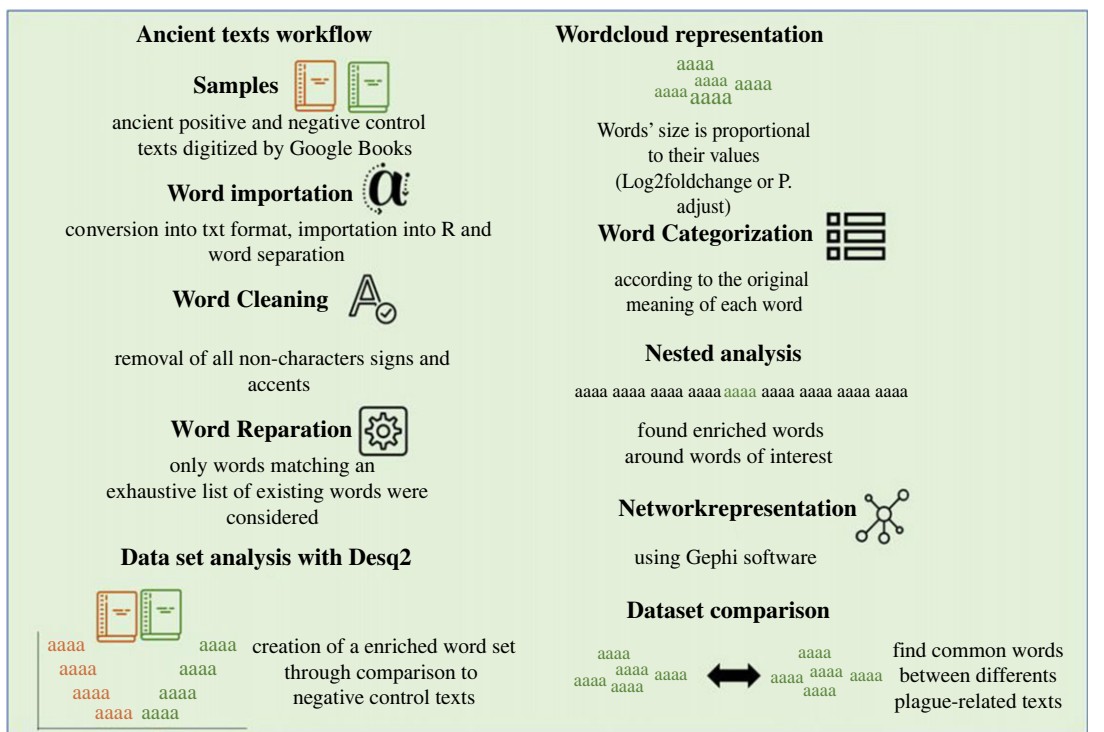

**Figure 1.** Workflow summarizing the main steps of the method developed in this study.

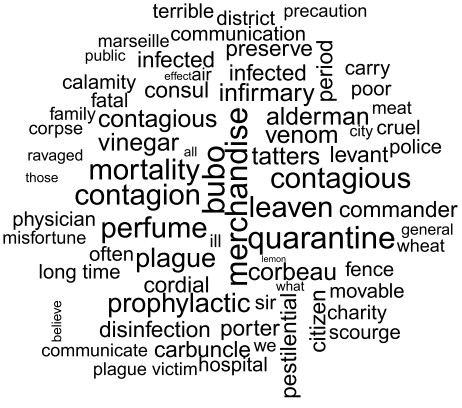

**Figure 2.** WordCloud displaying the words that were overrepresented in the Marseille plague-related texts ($p$-adj $\leq 0.05$ and $\log_2$-fold change greater than 0). The size of the words represents their adjusted $p$ enrichment in plague-related texts compared to negative control texts.

dysregulated genes in transcriptome analyses, we generated a list of 70 words ($\log_2$-fold change greater than 0 and $p$-adjusted (adj) $\leq 0.05$) (0.0007%), which were more overrepresented in plague-related texts than in control texts, as presented in WordCloud (figure 2; electronic supplementary material, table S1; data file S3). Two words were represented twice, i.e. 'contagious' and 'infected'. Our results confirmed the significance of two plague signs, namely 'bubo' ($p$-adj $= 5 \times 10^{-7}$) and 'carbuncle' ($p$-adj $= 0.003$). Notably, these two signs were continuously described, first by Procopius in antiquity [30], then by Guy de Chauliac in medieval times [31] and by Yersin [32] and Simond [33] at the beginning of the third and current plague pandemic. The significant detection of these two words, constituting internal positive controls, validated our method and allowed us to consider other enriched words significantly associated with the plague. Notably, the words 'rats' and 'fleas' appeared only 11 and 5 times in plague-related texts versus 5 and 3 times in negative control texts, respectively; the differences were not significant. Further careful examination revealed that the word 'rat' was misleading in five cases, resulting from the abbreviation of the word 'magistrate'. Ultimately, there

**Table 1.** Overrepresented or nested words identified as plague sources in the analysed records. Table representing the number of occurrences of each suspected plague source in plague-related texts from Marseille and northern Italy, featuring the percentage that each word is used by ancient authors as a source of plague. Words for the *bona fide* analysis were selected from and depended on the results of the nested analysis for their association with plague-related words.

| word | occurrence | number of times the term is used as a plague source | percentage (%) |
|---|---|---|---|
| overrepresented or nested words identified as plague sources during the Great Plague of Marseille, 1720–1722 | | | |
| merchandise | 810 | 561 | 69 |
| tatters | 225 | 196 | 87 |
| movable | 207 | 194 | 94 |
| clothes | 153 | 128 | 83 |
| air | 233 | 86 | 37 |
| meat | 133 | 8 | 6 |
| dog | 282 | 87 | 31 |
| overrepresented or nested words identified as plague sources during the plague epidemic of 1629–1631 in northern Italy | | | |
| stuff | 466 | 419 | 90 |
| feather | 29 | 26 | 90 |
| clothes | 89 | 55 | 62 |
| bed sheet | 25 | 15 | 60 |
| air | 180 | 98 | 54 |
| bed | 146 | 58 | 40 |

were six true occurrences in one text reporting dead rats in the streets and rats fleeing from houses where the plague was declared, but the author describes plague in general, and in any case, he does not mention rats during the plague of Marseille [34]. Likewise, the word 'flea' was mainly used to describe spots on the dura, observed during autopsy, that looked 'like flea bites' and only once to describe spots on the belly of a plague patient as 'like flea bites'. Only in one case did 'flea' unambiguously refer to the ectoparasite but this was in an out-of-plague context.

## 2.3. Categorizing the Marseille word database

By an analogy with bioinformatics, where clusters of orthologous genes (COGs) are defined as functional categories [35], the 70 plague words were classified into 19 categories of words, with each category comprising words related to the same function in the plague-related texts (electronic supplementary material, data file S4). The category of 'Other' was the most abundant, comprising 15/70 (21%) unclassified words, followed by the categories of 'Consequence' (10/70 words, 14%), 'Public Response' (8/70 words, 11%) and 'Nature of Plague' (7/70 words, 10%). Some words, i.e. 'police', 'physician', 'hospital' and 'infirmary', were classified into two different categories: 'Prevention' and 'Public Response'. Indeed, it appeared that these professions or places already existed before the plague surged due to the catastrophic situation, as demonstrated by the construction of additional infirmaries and hospitals and the increase in the numbers of doctors and police personnel. To address the assignment of some words to several categories, we specifically consolidated the category 'Sources' by analysing every sentence in which each word of this category was used to determine the number of its occurrence as a *bona fide* plague source. This analysis indicated that the word 'movable' was used in 194/207 (94%) occurrences as a plague source; 'tatters' was used in 196/225 (87%) occurrences as a plague source; 'clothes' was used in 128/153 (83%) occurrences as a plague source; and 'merchandise' was used in 561/810 (69%) occurrences as a plague source (table 1).

## 2.4. Marseille word database: nested and network analyses

A nested analysis was used to contextualize the 70 words overrepresented in the plague-related texts by extracting 25 of these overrepresented words downward and 25 upward and then testing

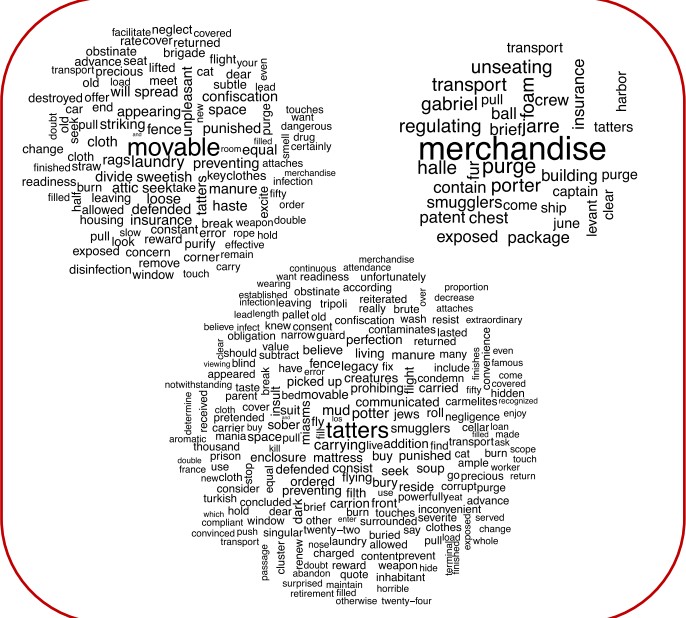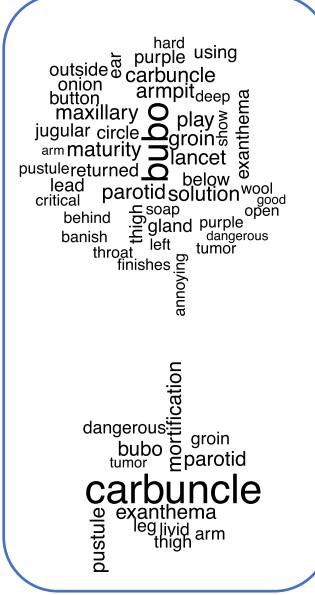

**Figure 3.** WordCloud representations of the nested analysis, displaying the overrepresented words (*p*-adj < 0.001) attached to the plague sources (movable, tatters and merchandise) (red panel, left) and plague symptoms (buboes and carbuncles) (blue panel, right) during the 1720–1722 plague of Marseille.

the significance of each. A WordCloud for each overrepresented word was generated using *p*-adj < 0.001 (electronic supplementary material, data file S5; figure S3). As an illustration of this nested analysis representation, the word 'merchandise' (figure 3) in the category of 'Source' was significantly associated with 'Levant', 'smugglers', 'June', 'fur', 'ship', 'clear', 'unseating', 'exposed', 'purged', 'tatters', 'transport', 'porter' and 'Jarre' (the name of a quarantine island in the Marseille gulf) (*p*-adj < 0.001) (figure 3). Likewise, in the 'Symptoms' category, the word 'carbuncles' (the gangrenous lesions consecutive to the blisters that correspond to a microbe's gateway through the skin) [33] had 625 occurrences and was significantly associated with 'groin', 'parotid', 'arms', 'leg' and 'thigh'; all words indicative of the topography of the lesions (*p*-adj < 0.001). The word 'bubo' (the lymph node proximal to the inoculation site (blister or carbuncle)) [33,36] had 948 occurrences and was significantly associated with 'groin', 'armpit', 'arm', 'throat', 'parotid', 'ear', 'maxillary', 'jugular' and 'thigh' (*p*-adj < 0.001) (figure 3). All words designating localizations were, unsurprisingly, similar to those associated with 'carbuncles', with 95% similarity. Concerning plague sources, the words 'tatters', 'movable' (*p*-adj < 0.001), 'cloth' and 'air' (*p* < 0.005) were found to be linked with 'infected' and multiple other plague-related words (figure 4). The word 'tatters' was associated with infection (e.g. 'kill', 'corrupt', 'communicated', 'filth', 'contaminates', 'miasms', 'infect', 'infection'; *p*-adj < 0.001), disinfection-related words (e.g. 'purge', 'burn', 'aromatic', 'confiscation', 'punished'; *p*-adj < 0.001), and other interesting words related to the dissemination of the plague and its origin (e.g. 'smugglers', 'Turkish', 'Tripoli'; *p*-adj < 0.001). The word 'movable' was linked to words from the same lexical field as 'tatters' (e.g. 'tatters', 'cloth', 'clothes'; *p*-adj < 0.001). It was also associated with infection (e.g. 'dangerous', 'infection'; *p*-adj < 0.001) and disinfection-related words (e.g. 'burn', 'purge', 'destroyed', 'disinfection', 'perfume', 'purify', 'confiscation'; *p*-adj < 0.001). All these associations suggest that contemporaries associated the terms tatters, clothes and movable with plague sources that communicated the plague across the city. Likewise, the word 'merchandise' was linked to multiple disinfection-related words (e.g. 'purge', 'quarantine'; *p*-adj < 0.001). The word 'air' was dichotomously associated with contagion (e.g. 'bad', 'corrupt', *p*-adj < 0.001; 'infected' and 'poisonous', *p*-adj < 0.005) and with purification (e.g. 'purify', 'pure'; *p*-adj < 0.001). In figure 4, it can be clearly seen that the word 'infected' is linked with the words 'tatter', 'cloth', 'movable', 'air' and 'merchandise' (*p*-adj < 0.01). Some other networks for potential sources of plague and their relation with other words are reported in electronic supplementary material, figures S5 and S6.

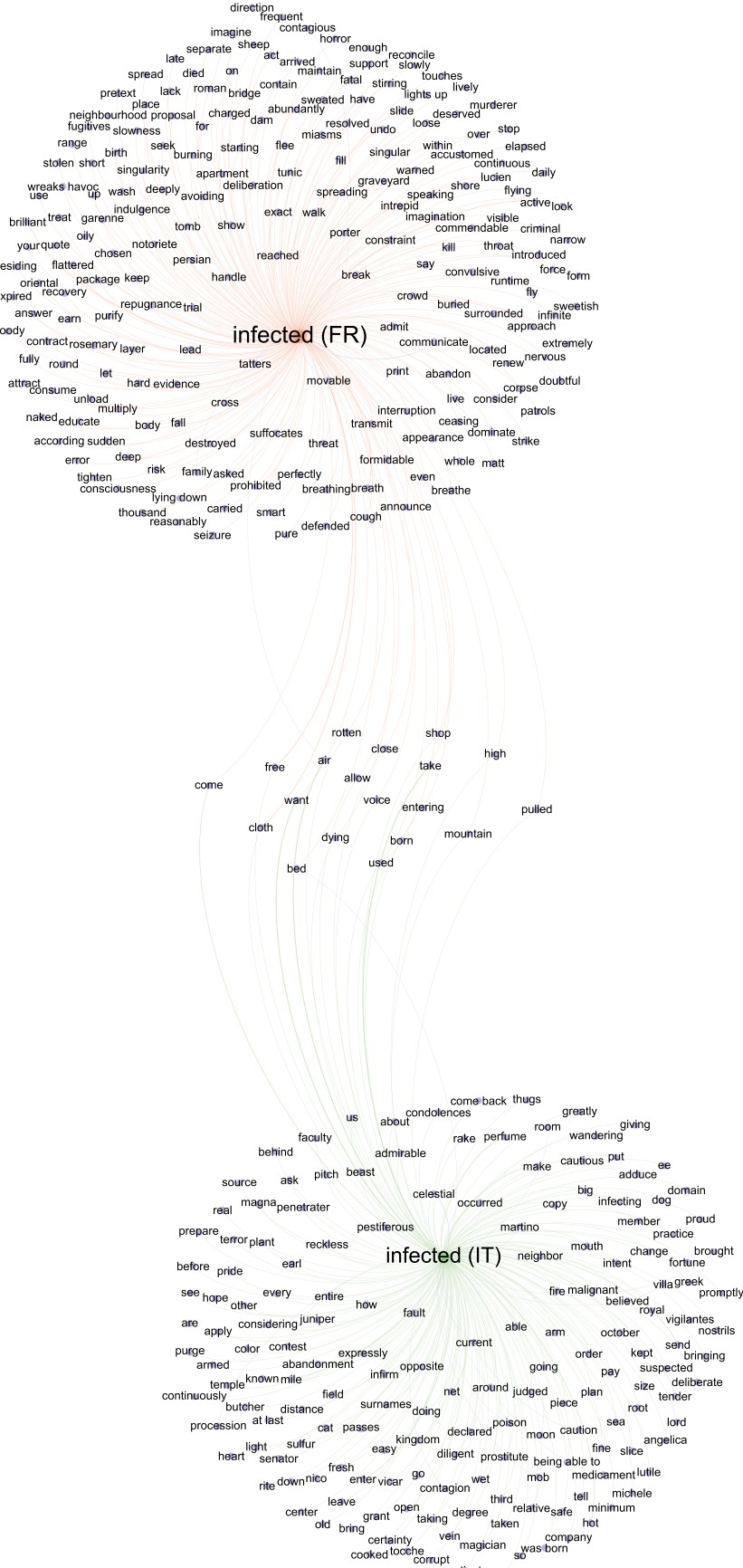

**Figure 4.** Network representation of the relations between words, generated using Gephi software and based on the adjusted *p*-values from the nested analysis (*p*-adj < 0.01) of the word 'infected' (in green for Marseille and in orange for northern Italy). Word label dimensions are proportional to the number of links with other words.

**Figure 5.** WordCloud displaying the words overrepresented in plague-related texts from northern Italy ($p$-adj $\leq$ 0.05 and log$_2$-fold change greater than 0). The size of the words represents their adjusted $p$ enrichment in plague-related texts compared to controls.

## 2.5. Documenting the 1629–1631 plague in northern Italy

We then processed 16 historical texts describing the 1629–1631 palaeomicrobiologically confirmed plague outbreak in northern Italy (electronic supplementary material, data file S6) [3,28]. Twenty Italian texts from the same historical period were used as negative control texts (electronic supplementary material, data file S7) (Jaccard index between two texts less than 0.03) (electronic supplementary material, figure S1B). The repertoire of words included 1 574 910, with a final word database of 1 063 483 total words, comprising 451 856 words from the plague-related texts and 611 627 words from the negative control texts. Using the DESeq2 package, we obtained a list of 147 overrepresented words (0.03%) ($p$-adj $\leq$ 0.05 and log$_2$-fold change greater than 0) (figure 5) (electronic supplementary material, table S2; figure S2B; data file S8). Seven words were represented twice, i.e. 'contagion', 'could', 'did', 'infected', 'our' and 'stuff', while 'were' was represented three times. These 147 words were then classified into the same categories used for the Marseille text analysis (electronic supplementary material, data file S4), and the words in the 'Source' category were consolidated as explained above (table 1). In particular, the Italian words 'roba' and 'robba', which have been translated into the English word 'stuff' and are used in a general way to indicate multiple terms such as movable, merchandise, food and especially clothes (electronic supplementary material, text S2) appeared more than 400 times in plague-related texts, and in 90% of the cases, the term was used by the author to indicate a plague source. Interestingly, the Italian words for 'rat' and 'mouse' were present only four times in the plague-related texts; twice in a figurative sense, once as a general premonition of plague (If the underground animals, which are worms, snakes, toads and mice, flee their dens to come upon the earth, unable to live amidst the extreme putrefaction of their mother, then, being on the ground, they die; they are the cause of infection) [37] and once in a statement that rats, together with dogs, cats, hens and doves, should be killed as a precaution to protect houses from the disease [37].

A nested analysis of Italian overrepresented words indicated that words such as 'clothes', 'beds', 'air' and 'rooms' were directly linked to infection, disinfection and danger-related words, indicating a clear association of them with potential plague sources (electronic supplementary material, data file S9; figure S4). In particular, the word 'bed' was linked to infection ('infected', 'contagion', 'pestiferous'; $p$-adj < 0.01), disinfection ('purge', 'perfume', 'air', 'boil'; $p$-adj < 0.01) and 'caution' ($p$-adj < 0.01). Similarly, the word 'cloth' was found to be associated with plague-related words, such as 'infected' ($p$-adj < 0.01), 'caution' ($p$-adj < 0.001) and 'poison' ($p$-adj < 0.01). The word 'poison' was also associated with 'clothes' ($p$-adj < 0.01). Moreover, the word 'air' was linked to 'infected' ($p$-adj < 0.001), 'bed' ($p$-adj < 0.001) and 'pestiferous' ($p$-adj < 0.01).

## 3. Plague pan-, core- and accessory-lexicomes

To define the plague pan-, core- and accessory-lexicomes (respectively, the complete collection of words associated with the plague, the plague words shared between the French and Italian lexicomes, and the

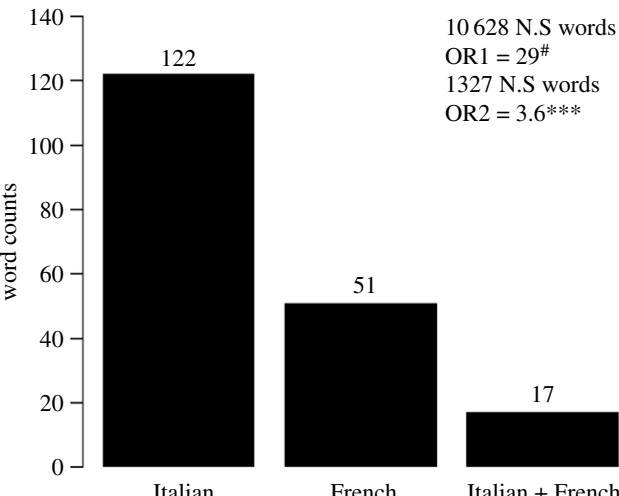

**Figure 6.** Barplot displaying the numbers of unique and significantly overrepresented words from Italian (IT) or French (FR) sources or shared between both. The overlap of significantly overrepresented words between Italian and French texts was tested using a Fisher test on all nonsignificant words (OR1) or nonsignificant words with a baseMean greater than the lower quartile of significant words (OR2). *p*-value < 0.001***; *p*-value < 0.00001#.

plague words specific to either episode), all the French and Italian words were translated into English using the translation function in Microsoft Word. Misleading translations of four words (three from the French texts and one from the Italian texts) resulted from either a truncated word (e.g. 'conta' for 'contamination' or 'contagion') or an isolated letter (e.g. 'o'), whereby a DESeq2 analysis incorporated a total of 68 unique words for Marseille and more than twice that number (139 unique words) for northern Italy. This difference in the richness of the repertoires of the two datasets and the stronger association between words obtained from the nested analysis of the French texts (electronic supplementary material, datasets S5 and S9) could not be related to the profession of the authors (i.e. physician and non-physician) (electronic supplementary material, datasets S1 and S6 and figure S7) but may possibly result from political and linguistic fragmentation of seventeenth century Italy, as opposed to the eighteenth century centralized France. Additionally, the Italian texts describe an epidemic covering multiple cities in multiple small countries in northern Italy, while the French texts cover a much more geographically limited region. The plague pan-lexicome was composed of 10 743 words and included a core-lexicome of 2645 words: 4063 words were specific to the Marseille plague and 4035 to the Italian plague. Regarding overrepresented unique words, there were 207 in the pan-lexicome (figure 6); the core-lexicome had 17 unique words. A total of 51 words specific to Marseille and 122 specific to northern Italy constituted the two accessory-lexicomes. The words that were significantly enriched in the Marseille plague texts had a more significant probability of being enriched in the texts related to the 1629–1631 plague in northern Italy than in the control texts (Fisher's exact test, performed with two conditions: *p*-value < 0.001***; *p*-value < 0.00001#). The French and Italian datasets were then graphically merged to create networks of common and unique words. The words of interest present in both datasets were compared and placed together to form a network that evidenced nested words that were unique or common to both datasets (*p*-adj < 0.01). The network representing the significant symptoms of plague (bubo and carbuncle) showed an overlap between the Marseille and northern Italy datasets corresponding to the most common locations for buboes and carbuncles (groin, armpit, throat, parotid, leg, and arm) (figure 7). Interestingly, three words, 'air', 'bed' and 'cloth', from the category of 'Sources', overlapped in the network for 'infected' (figure 4). No other potential plague sources were observed to be common between the Marseille and northern Italy datasets.

## 4. Discussion

The updated lexicographic analyses of ancient texts describing deadly outbreaks efficiently and effectively identified them as plague-describing texts and provided valuable information regarding the plague dynamics during the second pandemic.

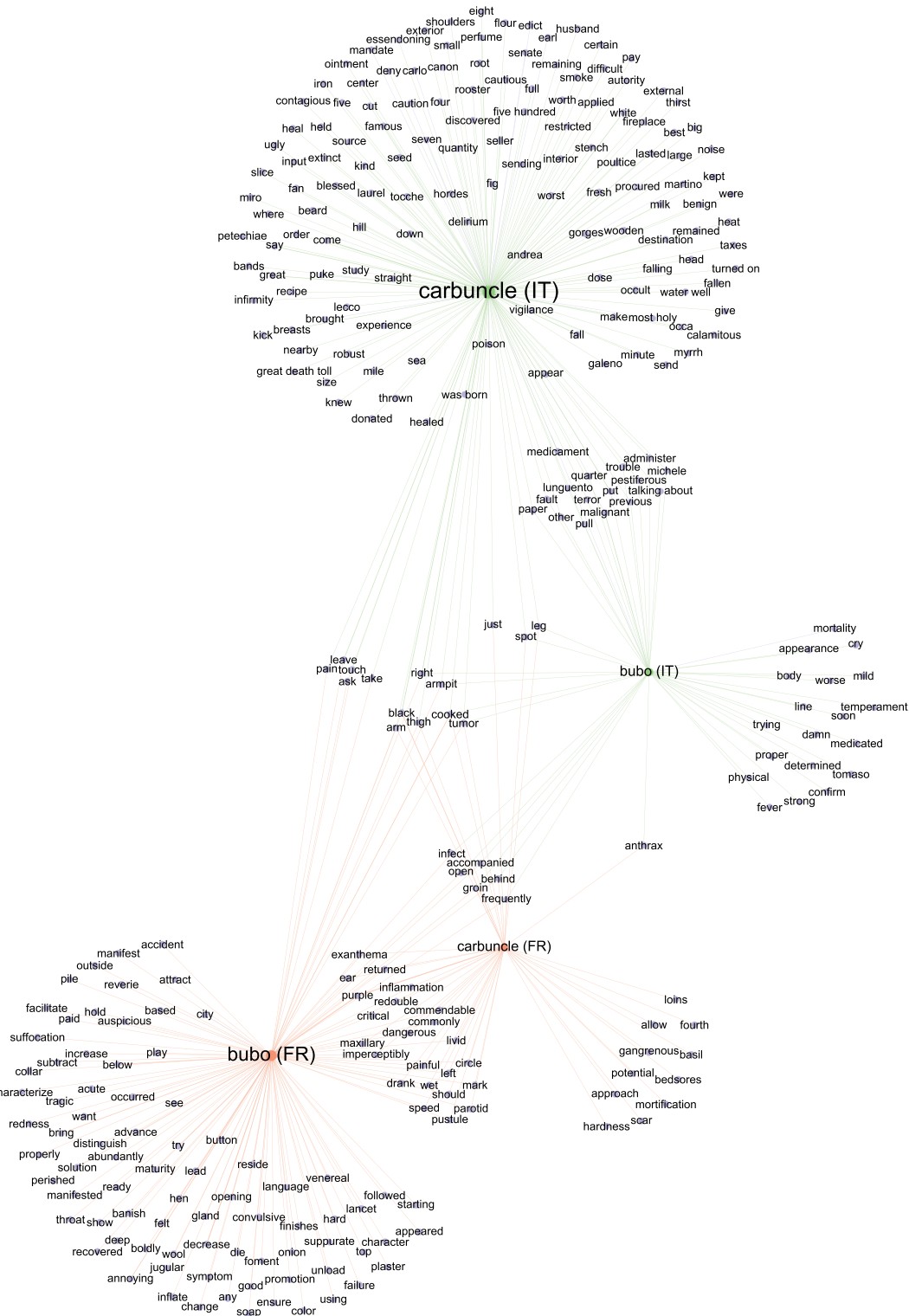

**Figure 7.** Word network representation of the words related to two plague symptoms, 'carbuncle' and 'bubo', from the Marseille plague (displayed in green) and the northern Italian plague (displayed in orange). Network generated with Gephi software, based on the adjusted *p*-values from the nested analysis (*p*-adj < 0.01). Word label dimensions are proportional to the number of links with other words.

We developed and used a method derived from automatic bioinformatics to obtain quantitative information from ancient texts. Our data, coupled with existing palaeomicrobiological analyses [3,4,8,26,28], confirm that these ancient texts describe a plague caused by *Y. pestis*, as revealed by the

significant presence of the words 'buboes' and 'carbuncles', which were the two main symptoms of the plague during the second pandemic. Furthermore, our analysis confirmed some extant research results, such as the role of clothes and textiles in the spread of plague [21,38]. One contribution of our analysis is that it expands the scope of possibilities concerning the sources and routes of contamination during two plague outbreaks. Contemporaries did not identify animals as sources of plague. The historical texts did not significantly mention animals as a source of plague, which disrupts the traditional depiction of plague as a primary zoonosis during the second pandemic [16]. In particular, our analysis failed to obtain 'rats' (or 'mice') and 'fleas' from ancient texts, an observation already articulated by Cohn [39]. These terms did not reach significance and were not enriched in plague-related texts, revising our understanding of the role played by rats and fleas in the transmission of the plague during the second pandemic. Controversially, the absence of rats from medieval texts could be attributed to two factors: a disinterest of the authors in pests and their linguistic inability to identify rats or mice in Latin [40]. Such linguistic issues seem to have been resolved in the modern era with the use of French and Italian; both 'rats' and 'mice' were used as distinct words in more recent texts [41–43], and one entire treatise in French was dedicated to the role played by rats throughout human history [44]. Although a study has shown that rats were probably involved in a human plague epidemic in Gdansk (Poland) during the fifteenth to sixteenth centuries [45], the impact of rats on the spread of massive plague outbreaks was probably trivial. There is an absence of rats in ancient plague texts and a scarcity of archaeozoological evidence, as reflected in extant mathematical models [15,16,24,39,46]. Concerning alternative sources of zoonotic plague, the word 'dog' was enriched in Marseille plague texts but never reached significance in the nested analysis. A careful reading of its 282 occurrences indicated that in 87 instances, dogs spread plague in Marseille by carrying infected tatters, clothes, dressings and bandages of plague victims, yet dogs were relatively resistant to plague, as demonstrated by contemporary experimental observations [47,48] and modern studies [49,50]. Other products derived from animal sources, such as meat or feathers, were also mentioned as possible plague sources, but a careful reading of texts disqualified the word 'meat', which was used as a remedy forming part of the broth to be given to plague victims. Likewise, feathers were actually used to form pillows or mattresses, and as our results have shown, bedding elements were viewed with suspicion and considered potential sources of the disease.

In the texts investigated here, word usage was in the context of the miasma and contagionist theories during the seventeenth to eighteenth centuries in Europe. Words such as 'air', 'leaven', 'pestilential', 'venom', 'pestilent', 'pestilence', 'poison' belonged to the miasma theory repertoire after its conceptualization in Hippocratic texts, inferring that plague was caused by corrupted vapours filling the air, called 'miasmas' [51] whereas words such as 'contagious', 'contagion', 'tatters', 'movable', 'stuff' belonged to the contagionist theory repertoire developed in the sixteenth century after scientists such as Girolamo Fracastoro inferred that diseases were transmitted by direct contacts, clothing or air [52]. Data gathered in the present analysis confirm that any text investigated here could mix elements from both theories (i.e. that plague arises from miasma and was spread by contagion [53]). Contemporaries traced plague outbreaks in Marseille and northern Italy to imported 'stuff', a general word encompassing the significantly enriched words 'movable', 'tatters', 'clothes' and 'merchandise'. During the seventeenth to eighteenth centuries in Europe, the plague and any other infectious disease was thought to be transmitted by textiles, according to contagionist theory [54–56]. Specific references were made to fur imported via the maritime route into Marseille, with a possibly unanticipated role played by smuggling and clothes brought by the French and German imperial soldiers into northern Italy during the war of Mantua [28,57–59]. All such clothing was in direct contact with plague-stricken people, and contemporaries insisted on the dangers that these clothes represented for spreading plague. In addition to discussing the Marseille and northern Italy outbreaks, numerous historical texts reported interhuman transmissions of plague through the transportation of movables, merchandise or clothes belonging to plague-stricken persons [57,60,61]. Accordingly, the people of those times identified infective potentiality in textiles and other merchandise that came from infected places or with which plague victims had been in contact, as reported by witnesses of the 1575–1576 plague in northern Italy [21]. Indeed, the trafficking of the clothes of deceased plague victims was a major problem in epidemic periods, given that in preindustrial Europe, wages were so low that buying clothes was a luxury that ordinary people could only afford a few times in their lives [62]. There are multiple direct accounts of episodes in which buying, selling and stealing clothes was reported as having triggered plague in homes, small villages or even cities [57,62,63]. In Milan and Marseille, the purging and sequestering of all the clothes and movable goods of plague victims were two primary prophylactic measures.

Regarding the symptoms, in texts from both Marseille and northern Italy, the words 'carbuncle' and 'bubo' were significantly enriched. Interestingly, the word 'carbuncle', referring to the skin ulcer that follows the introduction of *Y. pestis* through the skin [33], seems to be very representative of the second plague pandemic, whereas it is rarely used during the third pandemic and no longer reported in modern cases of plague. Notably, 'buboes' and 'carbuncles' were localized all over the body, including the face and the neck (specifically the ear, jugular, parotid gland and maxillary regions), localizations that do not support a role for fleas because rodent fleas are known to mainly bite the lower part of the body (legs and thighs) [46]. Rather, the use of these terms is compatible with a role for human ectoparasites, including body lice and human fleas, as vectors of *Y. pestis*, which could explain the rapid spread of the epidemics [64–67], which is supported by epidemiological [15,68], historical [69] and archaeological evidence [16].

The text mining method that structures our analysis was based on the model of the bioinformatics method of quantitative analysis of differential gene expression to limit bias and give a differential expression of words with the unprecedented introduction of negative control texts. Nevertheless, this method may not be exempt from biases, rendering data interpretation biases that it is necessary to be conscious of and that constitute major sources for future improvements. The quality of ancient text digitization can be extremely variable, with word loss ranging from 4.5% to 46.3% after word filtration (electronic supplementary material, tables S1 and S2), potentially eliding valuable details. The choice of controls was also decisive because it makes it possible to orient the results on several levels. In this work, we stopped at the first level; that is, we used control texts (including medical texts) to neutralize the common words that were not specific to plague at that time. A second level of analysis could have been done, for example, by using control texts referring to other epidemic diseases to determine whether words such as 'bed', 'clothes', 'tatters', 'merchandise' or 'air' were plague specific or were associated with any infectious diseases according to miasmatic theory. However, we did not find any digitalized documents contemporary to the two plague outbreaks analysed here. Finally, translating the words from their mother language into English detracts from their linguistic richness, often simplifying a relevant concept; for example, our analysis yielded 9.983 different words in French versus 6.707 in English and 9.909 in Italian versus 6.834 in English.

In summary, the method proposed here offers a new way to automatically analyse historical epidemics and historical events in general. In this era of digitalization, where historical data are becoming increasingly accessible each day, we propose a quantitative method that is able to rapidly analyse textual data by extracting not only a specific lexicome associated with a group of texts but also the relations and co-occurrence of words in these texts. This method is able to highlight interesting and potentially crucial pieces of information that can be used in parallel to close analyses of texts to help a reader interpret the dynamics of historical events.

# 5. Material and methods

## 5.1. Ancient texts

The texts were retrieved from Google Books, Archive.org and Buisante.parisdescartes using the following keyword combinations: plague & Marseille and plague & Italy, with specific period ranges, i.e. plague & Marseille & 1720–1820 or plague & Italy & 1629–1680. A total of 32 historical texts describing the 1720–1722 Great Plague of Marseille (16 complete books) and the 1629–1631 plague in northern Italy, also known as the Manzoni plague, after the famous novel written by Alessandro Manzoni in 1827 [70] (13 complete books and three book chapters), were retrieved (electronic supplementary material, data files S1 and S6). Additionally, we retrieved 11 contemporary negative control texts for Marseille and 20 contemporary negative control texts for northern Italy, all written during the seventeenth to eighteenth centuries and describing surgery, plants, universal history, industry and architecture; these texts were scanned within the framework of the 'Google Books' program, undertaken by Google and the Archives program (electronic supplementary material, data files S2 and S7). The Google Books format was used after we compared this format with the OCR software Wondershare PDFelement 6 Pro scanned format on the Plague_FR_02 from the Marseille Corpus text. After filtration, Google Books version yielded 51 984 words versus 51 418 words in Wondershare PDFelement 6 Pro. Careful examination of both versions indicated that most of the 'words' scanned by Wondershare PDFelement 6 Pro were isolated letters present in our dictionary as determinant possessives or verbs. For instance, Google Books yielded one word 'Madame' translated by Wondershare PDFelement 6 Pro into six

'words' 'M', 'A', 'D', 'A', 'M', 'E'. This example suggested that using Wondershare PDFelement 6 Pro would imply an extensive control of texts before Deseq2 analysis, conversely to the method we aimed to develop. This observation was in agreement with Google Books which was acknowledged to have one of the most efficient text digitization systems [71,72].

## 5.2. French texts' disposition

To filter the raw text words (see §5.4), we used a list comprising 336 631 French words, conjugated verbs and proper nouns, compiled from the Français-Gutenberg Dictionary. The symbol 'ſ' ('long S', which was used in most languages in Europe until the industrial revolution) was automatically substituted with an 's' only if a modern version of the relevant term with an 's' existed in the list of French words. To standardize the words, plural forms ending with an 's' or 'x' were converted to the singular form only if the singular word was present in the list of French words.

## 5.3. Italian texts' disposition

Italian texts (plague-related texts and controls) presented ambiguities that could lead to errors in an analysis. Indeed, most Italian books present, on the upper part of each page, the name of the book or chapter. Another problem of Italian books is the repetition of the last word of each page at the beginning of the next page. To avoid problems due to the overrepresentation of some words, these two types of repetition were manually corrected by removing the name of the book/chapter from each page, together with the last word of each page. As reported above, the symbol 'ſ' (long S) was automatically substituted with an 's'. However, we observed that OCR programs tend to recognize the 'ſ' symbol as an 'f' and not as an 's'. To avoid the loss of the Italian word 'peste' (plague) due to an OCR mistranscription, the word 'pefte' (no meaning) was changed to 'peste' in all the texts. Accordingly, the only words identified were those present in the list of Italian words (including singular and plural forms, conjugated verbs, proper names and surnames, for a total of 931 657 words). Of these words, 427 (32%) were discarded. Moreover, one-letter words were removed, and singular and plural forms of the words were unified, as much as possible, into one word by following Italian grammatical rules for plural formation (electronic supplementary material, text S3). Plague-related and control text word counts were compared using the DESeq2 R package with default parameters (v. 1.22.2).

## 5.4. Historical text mining analyses

After the raw texts were imported into R (see electronic supplementary material, data) in .txt format, all the words were separated and converted to lowercase; accents, empty cells and nonletter characters such as '?', '!', ',', '[. ]', '-', '_', '%', '\$', '€', '#', '\\', '+', '*', ':', ';', '>', '<', '§', '&', '•', '«', '»', '[{}]','"', '\' and ''' were removed. The number of occurrences was computed for each word in the 63 texts. Regarding the plague-related texts, 3 624 476 words were identified, with 2 049 566 words from the French texts and 1 574 910 words from the Italian texts. Given that we were not able to systematically repair ancient mistranscribed words, only words matching those in an exhaustive list of French words (including singular and plural forms and all conjugated verbs, for a total of 336 631 words) and an exhaustive list of French cities, towns and places or Italian words were considered. Of approximately 3.5 million words, 2 721 742 were conserved (75%), resulting in 25 659 and 44 475 unique words for the French and Italian texts, respectively. Then, we applied a cut-off value to remove all unique words occurring fewer than 5 times in the French and Italian texts before the DESeq2 analysis, resulting in 10 315 and 10 645 unique words for the French and Italian texts, respectively. Subsequently, the plague-related and control text word counts were compared using the DESeq2 R package with default parameters (v. 1.22.2). This approach identified 70 French and 147 Italian terms that were overrepresented in plague-related texts compared to the control texts ($p$-adj < 0.05 and $\log_2$-fold change greater than 0) and 213 French and 313 Italian words that appeared at least 4 times more in plague-related texts than in control texts ($\log_2$-fold change greater than 2) (electronic supplementary material, data files S3 and S8).

## 5.5. Nested analysis

In a second step, we used a 'nested' analysis to identify the words that lay in close proximity to overrepresented words in the plague-related texts. We extracted words located ±25 words apart from significant terms and compared them to the remaining part of the texts using DESeq2. Significant

words ($p$-adj ≤0.001 and log$_2$-fold change greater than 0) were plotted using the 'WordCloud' R package (v. 2.6) (electronic supplementary material, data files S5 and S9).

## 5.6. Word translation

All French and Italian words were translated into English using the translation function in Microsoft Word. Each overrepresented (only the words with $p < 0.05$ or log$_2$-fold change greater than 0) and nested (words associated with overrepresented words) word was controlled by the operators for mistranslation and then corrected. Words with no English translation were maintained as they were; this was the case for one French word, 'corbeau' (a person responsible for burying plague victims), and four Italian words, namely 'monatti' (people responsible for burying plague victims), 'espurgatori' (people responsible for stuff-purging), 'untori' (people accused of intentionally spreading the disease using 'special' ointments), 'caldiera' (a tool used to untie the cocoons of silkworms), with no direct English translation. Words exhibiting an 'out of context' translation were also manually corrected, such as 'carbone', 'tacchi' and 'unti' in Italian texts and 'preservatif' in French texts.

## 5.7. Network representation using Gephi

A graphical representation of the nested analysis results was performed using Gephi 0.9.2 [73]. This software was used exclusively for representation purposes of the data and statistics obtained by the nested analysis. The results of this analysis were converted into nodes and edges to generate the network. At each node, a word was obtained from the nested analysis, and the edges between two words represented the association between them. Only words with strong statistical significance ($p$-adj $< 0.01$) were represented. The spatial position of the words in the network has no statistical meaning. To maximize the efficiency of the comparison between the French and Italian datasets for the construction of the comparison networks, the words nested with each word of interest and the words the word of interest was nested with were used. Therefore, an edge was created each time a word of interest was associated with another word with a significance determined by $p$-adj $< 0.01$.

Data accessibility. All data needed to evaluate the conclusions in the paper are present in the paper and/or the electronic supplementary material [74].

Authors' contributions. R.B.: conceptualization, data curation, formal analysis, funding acquisition, investigation, methodology, resources, writing—original draft, writing—review and editing; R.N.: data curation, formal analysis, funding acquisition, investigation, methodology, resources, writing—original draft, writing—review and editing; S.E.: project administration, supervision, writing—original draft, writing—review and editing; M.S.: project administration, resources, supervision, validation, writing—original draft, writing—review and editing; D.R.: project administration, supervision, validation, writing—original draft, writing—review and editing; M.D.: conceptualization, data curation, methodology, project administration, resources, supervision, validation, writing—original draft, writing—review and editing. All authors gave final approval for publication and agreed to be held accountable for the work performed therein.

Competing interests. The authors declare no competing interests.

Funding. This work was supported by the French Government under the Investissements d'avenir (Investments for the Future) programme managed by the Agence Nationale de la Recherche (ANR) (reference: Méditerranée Infection 10-IAHU-03). This work was supported by the Région Le Sud (Provence Alpes Côte d'Azur) and European funding from FEDER IHU PRIMMI.

Acknowledgements. The authors acknowledge Pierre Pontarotti (IHU-CNRS) for fruitful discussions about text analysis.

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
