## [Peer Review File · Royal Society Open Science]

Review History

RSOS-210039.R0 (Original submission)

Review form: Reviewer 1

Is the manuscript scientifically sound in its present form?

Yes

Are the interpretations and conclusions justified by the results?

No

Is the language acceptable?

Yes

Do you have any ethical concerns with this paper?

No

Have you any concerns about statistical analyses in this paper?

No

Recommendation?

Major revision is needed (please make suggestions in comments)

Comments to the Author(s)

This is an interesting article on a topic which is, unfortunately, very timely. However, its findings are mostly confirmative of what we already knew. The authors refer to an old hypothesis about the historical plague having been caused by a haemorrhagic fever – but I am not aware of any recent study that makes such a claim. The problem that does remain, is that the old paradigm of plague transmission, rat-rat flea-human, does not appear to be compatible with the ease with which historical plague was able to spread. Does this article contribute to solve this point? I do not think so, given that the claims that it makes are based on texts which reflect a specific medical tradition, and not necessarily direct observation of plague-specific epidemiological features (see my point below). And yet, as a confirmative exercise, the article does remain interesting. The text analysis techniques that it applies are novel in this field of enquiry. But the findings must be placed more properly in the context of recent plague studies, particularly those based on historical sources (textual or otherwise), in order not to become misleading and/or to appear keen on discovering hot water.

Regarding the coverage of the historical literature, a gaping hole in the article is the imperfect coverage of the work by S.K. Cohn, who is undoubtedly the scholar who has done the best and most encompassing research on textual traditions concerning plague. Indeed, he has produced detailed comparative analyses of two kinds of texts: medieval chronicles of the plague and early modern medical treatises about how to fight plague. The books that I am referring to are the following:

S.K. Cohn, *The Black Death Transformed*, Arnold, London 2002

S.K. Cohn, *Cultures of Plague. Medical Thought at the End of the Renaissance*. Oxford: Oxford University Press 2010

Based on these books, I believe that the authors currently have an imperfect grasp of the textual traditions that lead to the use of certain words in later texts concerning plague, as well as of the medical traditions that explain certain views about the causes and the spread of plague that resisted until the Bacteriological Revolution of the nineteenth century.

Take for example the connection between rats and plague, a topic that the authors mention in their article (arguing correctly that rats rarely appear in the texts, save for when they are seen as a sign “announcing” the plague). Cohn’s works will lead the authors to realize that on this point there is a very ancient textual tradition, dating back to the immediate post-Black Death decades, that resists across centuries and still informs texts about the plague produced in 17th C Italy and in 18th C France...

p. 2, plague of 1629-31: note that the most updated study of this plague is Alfani and Bonetti, “A survival analysis of the last great European plagues: The case of Nonantola (Northern Italy) in 1630”, *Population Studies*, 73(1), 2019, pp. 101-118. This article also provides an in-depth analysis of the (probable) means of transmission of the plague in a European historical setting, which is basically along the lines proposed by the authors but which should be taken into account as it is based on yet another kind of external evidence: micro-demographic data from parish records and nominative record linkage.

p. 5, “All these associations suggest that contemporaries considered tatters, clothes, and movable as plague sources that communicated plague across the city”. Well, my point here is that this was not solely considered to be the cause of plague, but of the vast majority of infectious diseases given the “contagionist” theory which was predominant in the medical science of the time. Hence a finding like this does not tell us anything specific about plague (let alone plague transmission) or at least, it could not immediately and simply be presumed to do so. Again, the authors need to

have a good understanding of pre-nineteenth century medical theory in order to truly understand their data & findings, and right now I am quite dubious that they possess such understanding. A recent book which might help (but this is just as a starting point!) is Snowden, *Epidemics and Society: From the Black Death to the Present*, Yale University Press 2019. Note that Snowden is not really updated in his coverage of the plague, hence I am advising the authors to consider him solely as concerns the history of European medical views about epidemics

About the fact that the 1629-31 epidemic in north Italy “was never microbiologically confirmed as plague” (p.5) the authors should check Seguin-Orlando, A., et al., "No particular genomic features underpin the dramatic economic consequences of 17th century plague epidemics in Italy", *ISCIENCE*, December 2020. Although I do believe that there are some issues with Seguin-Orlando et al.'s article, it seems to me that it offers some microbiological evidence – which was not really much needed given that based on other material there is little doubt that the 1629-30 epidemic in Italy was caused by “bubonic plague”....

Again at p. 5, maybe I am missing something here, but should the fact that words as “could”, “did”, and “our” are “represented twice”, and “were” even thrice, should tell us something? I can't see how this is a meaningful finding...

The bottom line of my comments is the following: this article would become stronger, and not weaken, if it accepted the intrinsic limitations of the approach that it proposes. The results are confirmative, but this is not per se a bad thing – given that the methods are novel for the field, having an extra confirmation of what we already thought to know is not simply reassuring: it is useful...

Review form: Reviewer 2

Is the manuscript scientifically sound in its present form?

No

Are the interpretations and conclusions justified by the results?

No

Is the language acceptable?

No

Do you have any ethical concerns with this paper?

No

Have you any concerns about statistical analyses in this paper?

No

Recommendation?

Major revision is needed (please make suggestions in comments)

Comments to the Author(s)

Rémi and colleagues have presented a quantitative analysis of text-based documents associated with two historic plague outbreaks to search for consistent language that might inform on methods of disease transmission. Texts from the Great Plague of Marseille, 1720 – 1722, are used as “positive controls”, since *Y. pestis* DNA has been identified in human remains associated with the epidemic. A cursory comparison against contemporary texts from non-plague associated

episodes was made, though the results of this comparison are discussed only with reference to mention of rats and fleas. The authors then apply this model to texts describing a presumed plague epidemic in Northern Italy (1629 – 1631), which they state has not yet been confirmed to be a plague outbreak via molecular analysis (though I refer the authors to the recently published work by Orlando et al, <https://www.sciencedirect.com/science/article/pii/S2589004221003515>).

While the lexicometry approach is both a relevant and welcome addition to methods of plague analysis, the manuscript lacks a clear research statement. Are they attempting to use this model to lend credence to the idea that the plague of Northern Italy was indeed an outbreak of *Y. pestis*? Has that been contested? If so, an exploration of the literature presenting both sides of the debate should be presented. Currently the manuscript reads as though the authors' main goal is to use these methods as an independent line of support for plagues of the second pandemic (those post-Black Death in Europe) having been transmitted via the human louse as opposed to the rat-flea model, the latter of which is well established for plagues of the 20th century. The introduction states that rat-flea models or airborne transmission "may not have been sufficient to fuel the massive medieval mortality, pointing to the complementary role of human lice...". While discrepancies in disease severity between the historic and modern forms of plague are known, their natural extension to human louse transmission is odd and ignores other theories to explain differences in disease progression between the past and today. Differences in susceptibility, both in terms of population health (e.g., frailty, see the work of Sharon DeWitte) and possible genetic changes that are only just beginning to be explored, are deserving of mention, as both would influence the severity of a plague epidemic independent of rat population densities. Also, it should be clear that the majority of the discussants do not dispute the rat-flea model, where the human louse model of plague transmission is a hypothesis discussed by comparatively few.

That said, I find their approach unconvincing to address their main goal as I understand it. If rats and fleas were not suspected as vectors of infection based on the concepts of contagion understood by the writers, why would they be mentioned in the texts? More discussion is needed to explore the understanding of disease dissemination by the authors of the texts that were queried. I would assume that most doctors understood infectious disease to be carried by air (the miasma theory), where discussion of "stuffs" as vectors of contagion, as they have identified here, bears further contextualisation. Most importantly, the authors essentially argue that rat-flea transmission is a less likely model of plague dissemination since the texts they consulted apparently make no mention of rat mortality (save for one, which they argue is not applicable because the "author spoke of plague in general"... So why is this not valid?). Does that really support the idea that plague was not transmitted by rats? Is absence of evidence truly evidence of absence in this case?

Last, the manuscript is over-run with hyperbolic statements that weaken their main arguments. Line numbers did not render properly in my pdf, but here are a few examples:

1. "Unbiased lexicometry" in the title, and the statement, introduction paragraph 3: "we designed a purposefully naïve and unbiased method". No analytical method is completely absent of bias. In this case biases include observations of the writers, culturally-defined selection of words to describe them, method of translation from French/Italian to English (can true word meaning be obtained through use of modern translation tools for historic texts?), researchers' choice of word meaning assignment, among many others.
2. Discussion, paragraph 2: "These results resolve a years-long debate as to the etiology of plague in historical sources". Are the debates really that controversial given the revelations based on ancient DNA? I fail to see how their method has uniquely resolved anything. If the authors feel differently, the controversies should be explored. Conventional wisdom, including Wikipedia (https://en.wikipedia.org/wiki/1629%E2%80%931631_Italian_plague accessed May,

2021), state that the Italian plague of 1629 was likely caused by *Y. pestis*, transmitted in the bubonic form.

3. Their central conclusion: they challenge the assumption that “these episodes [plague epidemics] resulted from the reactivation of local plague foci (6) by suggesting human ectoparasites as vectors...”. After reading the manuscript, I see no strong support for the human ectoparasite theory of plague transmission.

In short, the approach is relevant, but the conclusions are too strong and lack support. Presentation of their analysis through transparent acknowledgement of biases and limitations of the method, along with a more balanced presentation of the ideas circulating to explain the differences between historical and modern plagues would make for a more balanced manuscript. A clear and attainable research statement should also be defined.

Review form: Reviewer 3

Is the manuscript scientifically sound in its present form?

No

Are the interpretations and conclusions justified by the results?

No

Is the language acceptable?

Yes

Do you have any ethical concerns with this paper?

No

Have you any concerns about statistical analyses in this paper?

No

Recommendation?

Major revision is needed (please make suggestions in comments)

Comments to the Author(s)

In their submitted manuscript “unbiased lexicometry analyses illuminate plague dynamics during the second pandemic”, Barbieri et al. use automated text mining methods along with statistical analyses on a number of texts describing two historical plague outbreaks as well as ‘negative controls’ to identify unique characteristics of plague texts and to draw conclusions about the epidemiology of historical plague outbreaks of the Second Pandemic in Europe.

The presented approach is unconventional but offers an interesting case study for the possibility of systematic, objective, and automated text mining of digitized historical texts. The analyses are well-documented, and the paper is written in a concise and comprehensible way. However, I have major concerns regarding the study design and the conclusions/interpretations of the results, described in detail below. Therefore, I suggest the send the manuscript back to the authors for major revisions.

Major revisions:

- 1) My main concern is the poor study design regarding the ‘negative controls’. Currently, the authors use only texts that are unrelated to plague as controls. This is highly problematic, since several of the significant differences in the ‘lexicome’ are likely not associated with plague (*Yersinia pestis*) in particular, but infectious diseases in general. One example is the overrepresentation of “bed” in plague texts which might as well be associated with disease in general. Therefore, it is crucial to include a second negative control with texts about non-plague epidemics, ideally with known causative agent (e.g., historical smallpox or cholera outbreaks). This will add another level of discriminatory power to their analyses, ideally yielding ‘lexicomes’ that are common for infectious diseases and distinct plague ‘lexicomes’.
- 2) The presented analyses show no overrepresentation of ‘rats’ or ‘fleas’ in plague-related texts, interpreted by the authors as supportive for a human ectoparasite model for plague transmission. Whereas *argumenta ex silentio* are generally problematic, this case is even worse, as it completely neglects the historical component and is even contradictory. Following the *argumentum ex silentio*, one could argue as well that the non-overrepresentation of the word ‘bacterium’ is indicative of a non-bacterial causative agent. With this, I just want to emphasize that these Early Modern texts cannot be interpreted without acknowledging the environment in which they were written, at a time when most infectious disease transmission mechanisms were not understood and the miasmatic theory was commonly accepted (see ‘air’, ‘perfume’, ‘miasma’ in the analyses). This interpretation is furthermore contradictory, as (human) ‘lice’ and ‘fleas’ are also not overrepresented. The mere overrepresentation of words like ‘clothes’ or ‘tatters’ are thus not convincing, as they could also emerge from a different, contemporary model of disease transmission related to or equally wrong as the miasmatic theory.
- 3) This leads to the last major concern: the general lack of engagement with medical history. A discussion of the results within the framework of medical history would of course lower the explanatory power on some aspects, as shown above. On the other hand though, the manuscript would gain not only solidity and soundness by giving treating the historical texts as such, but would also gain an additional perspective on contemporary ideas and theories related to epidemic diseases. With regard to this, also a comparison of ‘medical’ versus ‘lay’ texts on the respective plague outbreaks might be of interest.

Minor revisions:

- 1) In line 94, the authors report that 23.1% of words were excluded due to poor digitization quality. This is tremendous and should not be simply tolerated. I suggest to try re-digitizing the texts, perhaps with different tools, to achieve a better rate.
- 2) Several of the texts date decades later than the respective plague outbreak, in one case even around a century. This could be clearly stated in the manuscript, and ideally the analyses should be performed also solely on texts that were not written more than 10 years after the respective outbreak.
- 3) The manuscript obviously deals with non-English texts written on plague outbreaks in France and Italy. In order to allow historians and philologists to evaluate the results of this paper, I suggest adding a supplementary table presenting all overrepresented words in their original language. Also, words in word clouds should be presented in a searchable text or table format.
- 4) In Table 1, the words overrepresented/nested in plague texts and classified as sources include ‘meat’ and ‘dog’ (Marseille) as well as ‘feather’ (Northern Italy). This should be discussed in the manuscript, as this might offer interesting insights into contemporary ideas about disease transmission.
- 5) A recent publication was able to retrieve ancient plague DNA from rat remains associated with an outbreak of the second pandemic (Morozova et al. 2020). This should be at least mentioned in the manuscript, as this at least proves the involvement of rats in historical plague outbreaks.
- 6) Also, recent publications of ancient plague genomes associated with the outbreak in Northern Italy around 1630 are not cited: Guellil et al. 2020, San Procolo a Naturno; and Seguin-Orlando et al. 2021, Lariey-Puy-Saint-Pierre.

Decision letter (RSOS-210039.R0)

Dear Mr Drancourt

The Editors assigned to your paper RSOS-210039 "Unbiased lexicometry analyses illuminate plague dynamics during the second pandemic." have now received comments from reviewers and would like you to revise the paper in accordance with the reviewer comments and any comments from the Editors. Please note this decision does not guarantee eventual acceptance.

Please submit your revised manuscript and required files (see below) no later than 21 days from today's (ie 15-Jun-2021) date. Note: the ScholarOne system will 'lock' if submission of the revision is attempted 21 or more days after the deadline. If you do not think you will be able to meet this deadline please contact the editorial office immediately.

on behalf of Marta Kwiatkowska (Subject Editor)
openscience@royalsociety.org

Associate Editor Comments to Author:
Comments to the Author:

Please carefully read the extensive feedback you've received from the three referees and ensure that you respond equally carefully in your revised paper, and make sure that your changes or rebuttals are fully delineated in your response to referees document.

Reviewer comments to Author:

Reviewer: 1

Comments to the Author(s)

This is an interesting article on a topic which is, unfortunately, very timely. However, its findings are mostly confirmative of what we already knew. The authors refer to an old hypothesis about the historical plague having been caused by a haemorrhagic fever – but I am not aware of any recent study that makes such a claim. The problem that does remain, is that the old paradigm of plague transmission, rat-rat flea-human, does not appear to be compatible with the ease with which historical plague was able to spread. Does this article contribute to solve this point? I do not think so, given that the claims that it makes are based on texts which reflect a specific medical tradition, and not necessarily direct observation of plague-specific epidemiological features (see my point below). And yet, as a confirmative exercise, the article does remain interesting. The text analysis techniques that it applies are novel in this field of enquiry. But the findings must be placed more properly in the context of recent plague studies, particularly those based on historical sources (textual or otherwise), in order not to become misleading and/or to appear keen on discovering hot water.

Regarding the coverage of the historical literature, a gaping hole in the article is the imperfect coverage of the work by S.K. Cohn, who is undoubtedly the scholar who has done the best and most encompassing research on textual traditions concerning plague. Indeed, he has produced detailed comparative analyses of two kinds of texts: medieval chronicles of the plague and early modern medical treatises about how to fight plague. The books that I am referring to are the following:

S.K. Cohn, *The Black Death Transformed*, Arnold, London 2002

S.K. Cohn, *Cultures of Plague. Medical Thought at the End of the Renaissance*. Oxford: Oxford University Press 2010

Based on these books, I believe that the authors currently have an imperfect grasp of the textual traditions that lead to the use of certain words in later texts concerning plague, as well as of the medical traditions that explain certain views about the causes and the spread of plague that resisted until the Bacteriological Revolution of the nineteenth century.

Take for example the connection between rats and plague, a topic that the authors mention in their article (arguing correctly that rats rarely appear in the texts, save for when they are seen as a sign “announcing” the plague). Cohn’s works will lead the authors to realize that on this point there is a very ancient textual tradition, dating back to the immediate post-Black Death decades, that resists across centuries and still informs texts about the plague produced in 17th C Italy and in 18th C France...

p. 2, plague of 1629-31: note that the most updated study of this plague is Alfani and Bonetti, “A survival analysis of the last great European plagues: The case of Nonantola (Northern Italy) in 1630”, *Population Studies*, 73(1), 2019, pp. 101-118. This article also provides an in-depth analysis of the (probable) means of transmission of the plague in a European historical setting, which is basically along the lines proposed by the authors but which should be taken into account as it is based on yet another kind of external evidence: micro-demographic data from parish records and nominative record linkage.

p. 5, “All these associations suggest that contemporaries considered tatters, clothes, and movable as plague sources that communicated plague across the city”. Well, my point here is that this was not solely considered to be the cause of plague, but of the vast majority of infectious diseases given the “contagionist” theory which was predominant in the medical science of the time. Hence a finding like this does not tell us anything specific about plague (let alone plague transmission) or at least, it could not immediately and simply be presumed to do so. Again, the authors need to have a good understanding of pre-nineteenth century medical theory in order to truly understand their data & findings, and right now I am quite dubious that they possess such

understanding. A recent book which might help (but this is just as a starting point!) is Snowden, *Epidemics and Society: From the Black Death to the Present*, Yale University Press 2019. Note that Snowden is not really updated in his coverage of the plague, hence I am advising the authors to consider him solely as concerns the history of European medical views about epidemics

About the fact that the 1629-31 epidemic in north Italy “was never microbiologically confirmed as plague” (p.5) the authors should check Seguin-Orlando, A., et al., "No particular genomic features underpin the dramatic economic consequences of 17th century plague epidemics in Italy", *ISCIENCE*, December 2020. Although I do believe that there are some issues with Seguin-Orlando et al.'s article, it seems to me that it offers some microbiological evidence - which was not really much needed given that based on other material there is little doubt that the 1629-30 epidemic in Italy was caused by “bubonic plague”....

Again at p. 5, maybe I am missing something here, but should the fact that words as “could”, “did”, and “our” are “represented twice”, and “were” even thrice, should tell us something? I can't see how this is a meaningful finding...

The bottom line of my comments is the following: this article would become stronger, and not weaken, if it accepted the intrinsic limitations of the approach that it proposes. The results are confirmative, but this is not per se a bad thing - given that the methods are novel for the field, having an extra confirmation of what we already thought to know is not simply reassuring: it is useful...

Reviewer: 2

Comments to the Author(s)

Rémi and colleagues have presented a quantitative analysis of text-based documents associated with two historic plague outbreaks to search for consistent language that might inform on methods of disease transmission. Texts from the Great Plague of Marseille, 1720 - 1722, are used as “positive controls”, since *Y. pestis* DNA has been identified in human remains associated with the epidemic. A cursory comparison against contemporary texts from non-plague associated episodes was made, though the results of this comparison are discussed only with reference to mention of rats and fleas. The authors then apply this model to texts describing a presumed plague epidemic in Northern Italy (1629 - 1631), which they state has not yet been confirmed to be a plague outbreak via molecular analysis (though I refer the authors to the recently published work by Orlando et al, <https://www.sciencedirect.com/science/article/pii/S2589004221003515>).

While the lexicometry approach is both a relevant and welcome addition to methods of plague analysis, the manuscript lacks a clear research statement. Are they attempting to use this model to lend credence to the idea that the plague of Northern Italy was indeed an outbreak of *Y. pestis*? Has that been contested? If so, an exploration of the literature presenting both sides of the debate should be presented. Currently the manuscript reads as though the authors' main goal is to use these methods as an independent line of support for plagues of the second pandemic (those post-Black Death in Europe) having been transmitted via the human louse as opposed to the rat-flea model, the latter of which is well established for plagues of the 20th century. The introduction states that rat-flea models or airborne transmission “may not have been sufficient to fuel the massive medieval mortality, pointing to the complementary role of human lice...”.

While discrepancies in disease severity between the historic and modern forms of plague are known, their natural extension to human louse transmission is odd and ignores other theories to explain differences in disease progression between the past and today. Differences in susceptibility, both in terms of population health (e.g., frailty, see the work of Sharon DeWitte) and possible genetic changes that are only just beginning to be explored, are deserving of mention, as both would influence the severity of a plague epidemic independent of rat

population densities. Also, it should be clear that the majority of the discussants do not dispute the rat-flea model, where the human louse model of plague transmission is a hypothesis discussed by comparatively few.

That said, I find their approach unconvincing to address their main goal as I understand it. If rats and fleas were not suspected as vectors of infection based on the concepts of contagion understood by the writers, why would they be mentioned in the texts? More discussion is needed to explore the understanding of disease dissemination by the authors of the texts that were queried. I would assume that most doctors understood infectious disease to be carried by air (the miasma theory), where discussion of “stuffs” as vectors of contagion, as they have identified here, bears further contextualisation. Most importantly, the authors essentially argue that rat-flea transmission is a less likely model of plague dissemination since the texts they consulted apparently make no mention of rat mortality (save for one, which they argue is not applicable because the “author spoke of plague in general”... So why is this not valid?). Does that really support the idea that plague was not transmitted by rats? Is absence of evidence truly evidence of absence in this case?

Last, the manuscript is over-run with hyperbolic statements that weaken their main arguments. Line numbers did not render properly in my pdf, but here are a few examples:

1. “Unbiased lexicometry” in the title, and the statement, introduction paragraph 3: “we designed a purposefully naïve and unbiased method”. No analytical method is completely absent of bias. In this case biases include observations of the writers, culturally-defined selection of words to describe them, method of translation from French/Italian to English (can true word meaning be obtained through use of modern translation tools for historic texts?), researchers’ choice of word meaning assignment, among many others.

2. Discussion, paragraph 2: “These results resolve a years-long debate as to the etiology of plague in historical sources”. Are the debates really that controversial given the revelations based on ancient DNA? I fail to see how their method has uniquely resolved anything. If the authors feel differently, the controversies should be explored. Conventional wisdom, including Wikipedia (https://en.wikipedia.org/wiki/1629%E2%80%93931631_Italian_plague accessed May, 2021), state that the Italian plague of 1629 was likely caused by *Y. pestis*, transmitted in the bubonic form.

3. Their central conclusion: they challenge the assumption that “these episodes [plague epidemics] resulted from the reactivation of local plague foci (6) by suggesting human ectoparasites as vectors...”. After reading the manuscript, I see no strong support for the human ectoparasite theory of plague transmission.

In short, the approach is relevant, but the conclusions are too strong and lack support.

Presentation of their analysis through transparent acknowledgement of biases and limitations of the method, along with a more balanced presentation of the ideas circulating to explain the differences between historical and modern plagues would make for a more balanced manuscript. A clear and attainable research statement should also be defined.

Reviewer: 3

Comments to the Author(s)

In their submitted manuscript “unbiased lexicometry analyses illuminate plague dynamics during the second pandemic”, Barbieri et al. use automated text mining methods along with statistical analyses on a number of texts describing two historical plague outbreaks as well as ‘negative controls’ to identify unique characteristics of plague texts and to draw conclusions about the epidemiology of historical plague outbreaks of the Second Pandemic in Europe.

The presented approach is unconventional but offers an interesting case study for the possibility of systematic, objective, and automated text mining of digitized historical texts. The analyses are well-documented, and the paper is written in a concise and comprehensible way. However, I have major concerns regarding the study design and the conclusions/interpretations of the results, described in detail below. Therefore, I suggest the send the manuscript back to the authors for major revisions.

Major revisions:

- 1) My main concern is the poor study design regarding the 'negative controls'. Currently, the authors use only texts that are unrelated to plague as controls. This is highly problematic, since several of the significant differences in the 'lexicome' are likely not associated with plague (*Yersinia pestis*) in particular, but infectious diseases in general. One example is the overrepresentation of "bed" in plague texts which might as well be associated with disease in general. Therefore, it is crucial to include a second negative control with texts about non-plague epidemics, ideally with known causative agent (e.g., historical smallpox or cholera outbreaks). This will add another level of discriminatory power to their analyses, ideally yielding 'lexicomes' that are common for infectious diseases and distinct plague 'lexicomes'.
- 2) The presented analyses show no overrepresentation of 'rats' or 'fleas' in plague-related texts, interpreted by the authors as supportive for a human ectoparasite model for plague transmission. Whereas argumenta ex silentio are generally problematic, this case is even worse, as it completely neglects the historical component and is even contradictory. Following the argumentum ex silentio, one could argue as well that the non-overrepresentation of the word 'bacterium' is indicative of a non-bacterial causative agent. With this, I just want to emphasize that these Early Modern texts cannot be interpreted without acknowledging the environment in which they were written, at a time when most infectious disease transmission mechanisms were not understood and the miasmatic theory was commonly accepted (see 'air', 'perfume', 'miasma' in the analyses). This interpretation is furthermore contradictory, as (human) 'lice' and 'fleas' are also not overrepresented. The mere overrepresentation of words like 'clothes' or 'tatters' are thus not convincing, as they could also emerge from a different, contemporary model of disease transmission related to or equally wrong as the miasmatic theory.
- 3) This leads to the last major concern: the general lack of engagement with medical history. A discussion of the results within the framework of medical history would of course lower the explanatory power on some aspects, as shown above. On the other hand though, the manuscript would gain not only solidity and soundness by giving treating the historical texts as such, but would also gain an additional perspective on contemporary ideas and theories related to epidemic diseases. With regard to this, also a comparison of 'medical' versus 'lay' texts on the respective plague outbreaks might be of interest.

Minor revisions:

- 1) In line 94, the authors report that 23.1% of words were excluded due to poor digitization quality. This is tremendous and should not be simply tolerated. I suggest to try re-digitizing the texts, perhaps with different tools, to achieve a better rate.
- 2) Several of the texts date decades later than the respective plague outbreak, in one case even around a century. This could be clearly stated in the manuscript, and ideally the analyses should be performed also solely on texts that were not written more than 10 years after the respective outbreak.
- 3) The manuscript obviously deals with non-English texts written on plague outbreaks in France and Italy. In order to allow historians and philologists to evaluate the results of this paper, I suggest adding a supplementary table presenting all overrepresented words in their original language. Also, words in word clouds should be presented in a searchable text or table format.

- 4) In Table 1, the words overrepresented/nested in plague texts and classified as sources include 'meat' and 'dog' (Marseille) as well as 'feather' (Northern Italy). This should be discussed in the manuscript, as this might offer interesting insights into contemporary ideas about disease transmission.
- 5) A recent publication was able to retrieve ancient plague DNA from rat remains associated with an outbreak of the second pandemic (Morozova et al. 2020). This should be at least mentioned in the manuscript, as this at least proves the involvement of rats in historical plague outbreaks.
- 6) Also, recent publications of ancient plague genomes associated with the outbreak in Northern Italy around 1630 are not cited: Guellil et al. 2020, San Procolo a Naturno; and Seguin-Orlando et al. 2021, Lariey-Puy-Saint-Pierre.

===PREPARING YOUR MANUSCRIPT===

===PREPARING YOUR REVISION IN SCHOLARONE===

Author's Response to Decision Letter for (RSOS-210039.R0)

See Appendix A.

RSOS-210039.R1 (Revision)

Review form: Reviewer 1

Is the manuscript scientifically sound in its present form?

Yes

Are the interpretations and conclusions justified by the results?

Yes

Is the language acceptable?

No

Do you have any ethical concerns with this paper?

No

Have you any concerns about statistical analyses in this paper?

No

Recommendation?

Major revision is needed (please make suggestions in comments)

Comments to the Author(s)

The current version of this article is a significant improvement over the older one. The article is now more focused on the methodological contribution and avoids making over-bold statements. My main residual concern is with the quality of the text, for two reasons. First, the revision and the general refocusing have left behind some feature of the earlier version – for example, the abstract does not present the article as mostly a methodological contribution, but as a work on plague transmission: as was the case for the earlier version, but not for the current one. Secondly, the new parts present many issues with the English. In my revision I had started listing small things to amend, but quickly stopped as the problem is quite systematic. A thorough revision of the English by a native speaker or professional is badly needed before acceptance.

Apart from issues with the English, I have a list of relatively minor comments:

Line 57, “Readings of second plague pandemics descriptions”, amend to “Readings of second plague pandemic descriptions”

Line 62, “they may even considered”, amend to “they may even have to be considered” (or at least, this is what I think that the authors mean: there was clearly something missing in the original sentence. Also note that in the light of recent findings from paleo-biology, the authors’ statement in its current formulation might sound too strong)

Line 63, “could to be related”, amend to “could be related”. IMPORTANT: from this line on I will stop reporting typos and issues with the English – I will instead make the general point that this article still needs to be checked very carefully for issues with the English.

About the plague in Italy in 1629-31 and more generally, about the plague in seventeenth-century Europe, the authors might also check G. Alfani, “Plague in Seventeenth Century Europe and the Decline of Italy: an Epidemiological Hypothesis”, *European Review of Economic History*, 2013, 17, 408-430. Also note that sometimes in the current version of the article the 1629-1631 plague is associated to Milan only, while in others it is (correctly) referred to a broader area. Alfani’s work mentioned above offers the most updated analysis of the territorial coverage of different plagues in 17th C. Italy. Maybe refer to this plague as “the 1629-31 plague in northern Italy” throughout the article?

p. 11, “texts in the Northern Italy dataset were largely written by nonmedical

professionals such as religious figures, historians and economists”, first, I struggle with understanding what an “economist” would be in seventeenth-century Italy, and secondly, by checking Dataset S6 I did not find any author that I would describe as an “economist” ... Maybe the authors are thinking about something closer to a “philosopher”? (otherwise, I suggest to simply state “professionals such as religious figures or historians”).

This being said, as there are some “medical professionals” among the authors of the treatises included in the sample, I think that it would be good to know whether there are differences in the “richness of the repertoire” of Italian professionals and non-professionals working on plague.

This kind of test would grant (or not) credibility to the proposed explanation of the difference between Italy and France. If the test fails, I wonder whether the political (hence, linguistic as well) fragmentation of 17th C Italy, as opposed to 18th C France, offers another viable explanation?

About the role played by animals of various kind (dogs included) in plague transmission, it might be worth checking a recent contribution by Monica Green, which suggests that at the time of the medieval Black Death many animal species might have contributed to spread the disease: Green, “Taking ‘Pandemic’ Seriously: Making the Black Death Global.” In *Pandemic Disease in the Medieval world. Rethinking the Black Death*, Kalamazoo and Bradford: Arc Medieval Press, 2015, pp. 27–61

p. 15, “1576 Plague”, this is best referred to as the “1575-76 plague”.

Review form: Reviewer 2

Is the manuscript scientifically sound in its present form?

No

Are the interpretations and conclusions justified by the results?

No

Is the language acceptable?

Yes

Do you have any ethical concerns with this paper?

No

Have you any concerns about statistical analyses in this paper?

No

Recommendation?

Accept with minor revision (please list in comments)

Comments to the Author(s)

I thank the authors for submission of the revised text. In light of their edits I am willing to recommend the manuscript for publication with the exception of two necessary changes:

1. They remove the sentence in the abstract “.... while no association was found with rats and fleas, suggesting that during the second plague pandemic human ectoparasites were probably major drivers of plague.”
2. Discussion, fourth last paragraph: “Our results confirmed the features already observed during the second pandemic that plague transmission was probably predominantly human-to-human [15,16,22], probably via human ectoparasites present in clothes”

The methods employed by the authors do not support these conclusions. As reviewer 1 has suggested, the value of the manuscript is sufficient as a novel methodological approach without the need for conclusions that go beyond the results. That said, a clearer research statement or motivation is still needed, especially since the results are confirmatory of those that have surfaced from other types of data analysis. The authors are of course welcome to describe the human ectoparasite theory as one of the circulating hypotheses on historical plague transmission, but they should not state that the model is supported by their analysis here.

Greater discussion of medical knowledge of the time and how this might affect the words chosen by the authors of the historical text is still needed. The miasma theory is only obliquely mentioned in the second last discussion paragraph, and should receive greater attention.

If Google Books was better than OCR based on tests they performed, the results of these tests should be disclosed.

Review form: Reviewer 3

Is the manuscript scientifically sound in its present form?

Yes

Are the interpretations and conclusions justified by the results?

Yes

Is the language acceptable?

Yes

Do you have any ethical concerns with this paper?

No

Have you any concerns about statistical analyses in this paper?

No

Recommendation?

Accept with minor revision (please list in comments)

Comments to the Author(s)

Although I'm not personally convinced regarding the strength of this approach to investigate alternative transmission hypotheses, I acknowledge that the authors toned down their claims, discuss the problems of their approach sufficiently, and backed up their argumentation with references. However, in this context, I cannot follow a change in the abstract which goes rather in the opposite direction. Therefore I request a minor revision in the following sentence:

"Moreover, plague-related words were associated with the words "merchandise", "movable", "tatters", "bed" and "clothes", while no association was found with rats and fleas, suggesting that during the second plague pandemic, human ectoparasites were probably major drivers of plague."

Please change this back to the original "These results support the hypothesis that during the second plague pandemic ..." while maintaining the newly introduced "probably". This reflects much better that the presented study is able to support this pre-existing hypothesis, but cannot suggest this hypothesis on its own.

Finally, I appreciate that the authors took my previous comments seriously and introduced additional data and discussions which increased the significance of this paper beyond the controversial transmission topic.

Decision letter (RSOS-210039.R1)

Dear Mr Drancourt

The Editors assigned to your paper RSOS-210039.R1 "Differential word expression analyses highlight plague dynamics during the second pandemic." have now received comments from reviewers and would like you to revise the paper in accordance with the reviewer comments and any comments from the Editors. Please note this decision does not guarantee eventual acceptance.

Please submit your revised manuscript and required files (see below) no later than 21 days from today's (ie 17-Sep-2021) date. Note: the ScholarOne system will 'lock' if submission of the revision is attempted 21 or more days after the deadline. If you do not think you will be able to meet this deadline please contact the editorial office immediately.

on behalf of Marta Kwiatkowska (Subject Editor)
openscience@royalsociety.org

Associate Editor Comments to Author:

Comments to the Author:

In the reviewers' eyes, the paper is much improved and we commend your efforts to address the concerns raised earlier; however, a number of matters remain to be resolved. These appear to be largely linguistic or relating to the authors' choice of phrasing for some parts of the paper (see the reviewer comments for details). We would like you to seek professional language editing support before resubmitting, and making sure you also take into account - and modify as needed - your manuscript to address the remaining issues identified. Please note that the Royal Society has a list of language editing services at <https://royalsociety.org/journals/authors/benefits/language-editing/> - some of which offer discounts to Society authors. Good luck in making these remaining modifications.

Reviewer comments to Author:

Reviewer: 1

Comments to the Author(s)

The current version of this article is a significant improvement over the older one. The article is now more focused on the methodological contribution and avoids making over-bold statements. My main residual concern is with the quality of the text, for two reasons. First, the revision and the general refocusing have left behind some feature of the earlier version - for example, the abstract does not present the article as mostly a methodological contribution, but as a work on plague transmission: as was the case for the earlier version, but not for the current one. Secondly, the new parts present many issues with the English. In my revision I had started listing small things to amend, but quickly stopped as the problem is quite systematic. A thorough revision of the English by a native speaker or professional is badly needed before acceptance.

Apart from issues with the English, I have a list of relatively minor comments:

Line 57, "Readings of second plague pandemics descriptions", amend to "Readings of second plague pandemic descriptions"

Line 62, "they may even considered", amend to "they may even have to be considered" (or at least, this is what I think that the authors mean: there was clearly something missing in the original sentence. Also note that in the light of recent findings from paleo-biology, the authors' statement in its current formulation might sound too strong)

Line 63, "could to be related", amend to "could be related". IMPORTANT: from this line on I will stop reporting typos and issues with the English - I will instead make the general point that this article still needs to be checked very carefully for issues with the English.

About the plague in Italy in 1629-31 and more generally, about the plague in seventeenth-century Europe, the authors might also check G. Alfani, "Plague in Seventeenth Century Europe and the Decline of Italy: an Epidemiological Hypothesis", *European Review of Economic History*, 2013, 17, 408-430. Also note that sometimes in the current version of the article the 1629-1631 plague is associated to Milan only, while in others it is (correctly) referred to a broader area. Alfani's work mentioned above offers the most updated analysis of the territorial coverage of different plagues in 17th C. Italy. Maybe refer to this plague as "the 1629-31 plague in northern Italy" throughout the article?

p. 11, "texts in the Northern Italy dataset were largely written by nonmedical professionals such as religious figures, historians and economists", first, I struggle with understanding what an "economist" would be in seventeenth-century Italy, and secondly, by checking Dataset S6 I did not find any author that I would describe as an "economist"... Maybe the authors are thinking about something closer to a "philosopher"? (otherwise, I suggest to simply state "professionals such as religious figures or historians").

This being said, as there are some "medical professionals" among the authors of the treatises included in the sample, I think that it would be good to know whether there are differences in the "richness of the repertoire" of Italian professionals and non-professionals working on plague. This kind of test would grant (or not) credibility to the proposed explanation of the difference

between Italy and France. If the test fails, I wonder whether the political (hence, linguistic as well) fragmentation of 17th C Italy, as opposed to 18th C France, offers another viable explanation? About the role played by animals of various kind (dogs included) in plague transmission, it might be worth checking a recent contribution by Monica Green, which suggests that at the time of the medieval Black Death many animal species might have contributed to spread the disease: Green, "Taking 'Pandemic' Seriously: Making the Black Death Global." In *Pandemic Disease in the Medieval World. Rethinking the Black Death*, Kalamazoo and Bradford: Arc Medieval Press, 2015, pp. 27-61
p. 15, "1576 Plague", this is best referred to as the "1575-76 plague".

Reviewer: 3

Comments to the Author(s)

Although I'm not personally convinced regarding the strength of this approach to investigate alternative transmission hypotheses, I acknowledge that the authors toned down their claims, discuss the problems of their approach sufficiently, and backed up their argumentation with references. However, in this context, I cannot follow a change in the abstract which goes rather in the opposite direction. Therefore I request a minor revision in the following sentence:

"Moreover, plague-related words were associated with the words "merchandise", "movable", "tatters", "bed" and "clothes", while no association was found with rats and fleas, suggesting that during the second plague pandemic, human ectoparasites were probably major drivers of plague."

Please change this back to the original "These results support the hypothesis that during the second plague pandemic ..." while maintaining the newly introduced "probably". This reflects much better that the presented study is able to support this pre-existing hypothesis, but cannot suggest this hypothesis on its own.

Finally, I appreciate that the authors took my previous comments seriously and introduced additional data and discussions which increased the significance of this paper beyond the controversial transmission topic.

Reviewer: 2

Comments to the Author(s)

I thank the authors for submission of the revised text. In light of their edits I am willing to recommend the manuscript for publication with the exception of two necessary changes:

1. They remove the sentence in the abstract "... while no association was found with rats and fleas, suggesting that during the second plague pandemic human ectoparasites were probably major drivers of plague."
2. Discussion, fourth last paragraph: "Our results confirmed the features already observed during the second pandemic that plague transmission was probably predominantly human-to-human [15,16,22], probably via human ectoparasites present in clothes"

The methods employed by the authors do not support these conclusions. As reviewer 1 has suggested, the value of the manuscript is sufficient as a novel methodological approach without the need for conclusions that go beyond the results. That said, a clearer research statement or motivation is still needed, especially since the results are confirmatory of those that have surfaced from other types of data analysis. The authors are of course welcome to describe the human ectoparasite theory as one of the circulating hypotheses on historical plague transmission, but they should not state that the model is supported by their analysis here.

Greater discussion of medical knowledge of the time and how this might affect the words chosen by the authors of the historical text is still needed. The miasma theory is only obliquely mentioned in the second last discussion paragraph, and should receive greater attention.

If Google Books was better than OCR based on tests they performed, the results of these tests should be disclosed.

===PREPARING YOUR MANUSCRIPT===

===PREPARING YOUR REVISION IN SCHOLARONE===

Author's Response to Decision Letter for (RSOS-210039.R1)

See Appendix B.

RSOS-210039.R2

Review form: Reviewer 1

Is the manuscript scientifically sound in its present form?

Yes

Are the interpretations and conclusions justified by the results?

Yes

Is the language acceptable?

Yes

Do you have any ethical concerns with this paper?

No

Have you any concerns about statistical analyses in this paper?

No

Recommendation?

Accept as is

Comments to the Author(s)

I think that the authors did a very good job in making this article the best that it could be. I think that it is now publishable as it is.

Review form: Reviewer 2

Is the manuscript scientifically sound in its present form?

Yes

Are the interpretations and conclusions justified by the results?

Yes

Is the language acceptable?

Yes

Do you have any ethical concerns with this paper?

No

Have you any concerns about statistical analyses in this paper?

No

Recommendation?

Accept with minor revision (please list in comments)

Comments to the Author(s)

I thank the authors for submission of their revised manuscript. I have two remaining requests.

1. The authors should amend the statement in the last sentence of their abstract from "... including the sources for the causative *Yersinia pestis*." to something along the lines of "which can inform on the potential sources for the causative *Yersinia pestis*." It is important that the authors apply language to indicate that hypotheses are being developed or further supported by their analytical model, but since their methods offer indirect support with known biases that they now explain, they cannot unequivocally settle outstanding topics of debate such as the source and method of transmission of *Y. pestis* in the second pandemic.

2. Line 343 "Rather, the use of these terms suggests a role of for human ectoparasites, including body lice and human fleas, as vectors of *Y. pestis*..." should be changed to: "Rather, the use of these terms is compatible with a role for ectoparasites human ectoparasites, including body lice and human fleas, as vectors of *Y. pestis*...". I remain unconvinced that their analysis here provides any further evidence to support the human ectoparasite theory of plague dissemination, and qualifying language in discussion of this point is needed to make reasonable doubt clear.

Review form: Reviewer 3

Is the manuscript scientifically sound in its present form?

Yes

Are the interpretations and conclusions justified by the results?

Yes

Is the language acceptable?

Yes

Do you have any ethical concerns with this paper?

Yes

Have you any concerns about statistical analyses in this paper?

No

Recommendation?

Accept as is

Comments to the Author(s)

I think the paper improved significantly during the review process, most importantly by focusing on the methodology.

Decision letter (RSOS-210039.R2)

Dear Mr Drancourt

On behalf of the Editors, we are pleased to inform you that your Manuscript RSOS-210039.R2 "Differential word expression analyses highlight plague dynamics during the second pandemic." has been accepted for publication in Royal Society Open Science subject to minor revision in accordance with the referees' reports. Please find the referees' comments along with any feedback from the Editors below my signature.

Please submit your revised manuscript and required files (see below) no later than 7 days from today's (ie 18-Nov-2021) date. Note: the ScholarOne system will 'lock' if submission of the revision is attempted 7 or more days after the deadline. If you do not think you will be able to meet this deadline please contact the editorial office immediately.

on behalf of Prof Marta Kwiatkowska (Subject Editor)
openscience@royalsociety.org

Associate Editor Comments to Author:

Please make the final amendments requested by one of the reviewers before resubmitting for final consideration.

Reviewer comments to Author:

Reviewer: 3

Comments to the Author(s)

I think the paper improved significantly during the review process, most importantly by focusing on the methodology.

Reviewer: 1

Comments to the Author(s)

I think that the authors did a very good job in making this article the best that it could be. I think that it is now publishable as it is.

Reviewer: 2

Comments to the Author(s)

I thank the authors for submission of their revised manuscript. I have two remaining requests.

1. The authors should amend the statement in the last sentence of their abstract from "... including the sources for the causative *Yersinia pestis*." to something along the lines of "which can inform on the potential sources for the causative *Yersinia pestis*." It is important that the authors apply language to indicate that hypotheses are being developed or further supported by their analytical model, but since their methods offer indirect support with known biases that they now explain, they cannot unequivocally settle outstanding topics of debate such as the source and method of transmission of *Y. pestis* in the second pandemic.

2. Line 343 "Rather, the use of these terms suggests a role of for human ectoparasites, including body lice and human fleas, as vectors of *Y. pestis*..." should be changed to: "Rather, the use of these terms is compatible with a role for ectoparasites human ectoparasites, including body lice and human fleas, as vectors of *Y. pestis*...". I remain unconvinced that their analysis here provides any further evidence to support the human ectoparasite theory of plague dissemination, and qualifying language in discussion of this point is needed to make reasonable doubt clear.

===PREPARING YOUR MANUSCRIPT===

one version should clearly identify all the changes that have been made (for instance, in coloured highlight, in bold text, or tracked changes);

===PREPARING YOUR REVISION IN SCHOLARONE===

To revise your manuscript, log into <https://mc.manuscriptcentral.com/rsos> and enter your Author Centre - this may be accessed by clicking on "Author" in the dark toolbar at the top of the

page (just below the journal name). You will find your manuscript listed under "Manuscripts with Decisions". Under "Actions", click on "Create a Revision".

-- If you are requesting an article processing charge waiver, you must select the relevant waiver option (if requesting a discretionary waiver, the form should have been uploaded, see 'File upload' above).

-- If you have uploaded any electronic supplementary (ESM) files, please ensure you follow the guidance at <https://royalsociety.org/journals/authors/author-guidelines/#supplementary-material> to include a suitable title and informative caption. An example of appropriate titling and captioning may be found at https://figshare.com/articles/Table_S2_from_Is_there_a_trade-off_between_peak_performance_and_performance_breadth_across_temperatures_for_aerobic_scope_in_teleost_fishes_/3843624.

At the 'Review & submit' step, you must view the PDF proof of the manuscript before you will be able to submit the revision. Note: if any parts of the electronic submission form have not been

completed, these will be noted by red message boxes - you will need to resolve these errors before you can submit the revision.

Author's Response to Decision Letter for (RSOS-210039.R2)

See Appendix C.

Decision letter (RSOS-210039.R3)

Dear Mr Drancourt,

I am pleased to inform you that your manuscript entitled "Differential word expression analyses highlight plague dynamics during the second pandemic." is now accepted for publication in Royal Society Open Science.

on behalf of Prof Marta Kwiatkowska (Subject Editor)
openscience@royalsociety.org

Appendix A

Dear Editor,

Please find enclosed the revised version of our manuscript RSOS-210039 "Differential word expression analyses highlight plague dynamics during the second pandemics." by Barbieri R. and collaborators, along with the answers of the authors to the Editor and Reviewers' comments.

Reviewer comments to Author:

Reviewer: 1

Comments to the Author(s)

This is an interesting article on a topic which is, unfortunately, very timely. However, its findings are mostly confirmative of what we already knew. The authors refer to an old hypothesis about the historical plague having been caused by a haemorrhagic fever – but I am not aware of any recent study that makes such a claim.

Authors' answer: The authors acknowledge this general, positive comment. That the work is referred by the reviewer as mostly "confirmative" is a positive comment, confirming the validity of the method here reported; the method being the main message of this work. Then, indeed the particular topic of plague has been chosen as a positive control situation on which to build the original reading process here reported which is, once again, the original contribution of the work. Also, the authors do agree with the reviewer that referring to haemorrhagic fever was inappropriate in the context of cumulative evidence for *Yersinia pestis* and, accordingly, the sentence has been removed (Line 55).

The problem that does remain, is that the old paradigm of plague transmission, rat-rat flea-human, does not appear to be compatible with the ease with which historical plague was able to spread. Does this article contribute to solve this point? I do not think so, given that the claims that it makes are based on texts which reflect a specific medical tradition, and not necessarily direct observation of plague-specific epidemiological features (see my point below). And yet, as a confirmative exercise, the article does remain interesting. The text analysis techniques that it applies are novel in this field of enquiry. But the findings must be placed more properly in the context of recent plague studies, particularly those based on historical sources (textual or otherwise), in order not to become misleading and/or to appear keen on discovering hot water.

Authors' answer: The authors acknowledge this remark and the given opportunity to clarify the contribution of the work in the field of history. As it was obvious for the authors (and clearly the reviewer with who the authors agree on that point) that any observation is made in one particular historical context, precisely the authors introduced an original method of reading ancient texts, making case of the unprecedented variable of negative control texts, to neutralize as far as possible the cultural bias. This point is clarified in the revised version (Lines 339-357). Furthermore, the authors fully agree with the reviewer that enthusiasm about efficiency of the method may have led to some over interpretations of data; this has been corrected all along the revised version.

Regarding the coverage of the historical literature, a gaping hole in the article is the imperfect coverage of the work by S.K. Cohn, who is undoubtedly the scholar who has done the best and most encompassing research on textual traditions concerning plague. Indeed, he has produced detailed comparative analyses of two kinds of texts: medieval chronicles of the plague and early modern medical treatises about how to fight plague. The books that I am referring to are the following:

S.K. Cohn, *The Black Death Transformed*, Arnold, London 2002

S.K. Cohn, *Cultures of Plague. Medical Thought at the End of the Renaissance*. Oxford: Oxford University Press 2010

Based on these books, I believe that the authors currently have an imperfect grasp of the textual traditions that lead to the use of certain words in later texts concerning plague, as well as of the medical traditions that explain certain views about the causes and the spread of plague that resisted until the Bacteriological Revolution of the nineteenth century.

Take for example the connection between rats and plague, a topic that the authors mention in their article (arguing correctly that rats rarely appear in the texts, save for when they are seen as a sign "announcing" the plague). Cohn's works will lead the authors to realize that on this point there is a very ancient textual tradition, dating back to the immediate post-Black Death decades, that resists across centuries and still informs texts about the plague produced in 17th C Italy and in 18th C France...

Authors' answer: The reviewer is perfectly right and, accordingly, the authors now present complementarity of data yielded by the method of reading they invented, with the data

yielded by the tremendous works by Cohn and his collaborators. Accordingly, references by Cohn and collaborators are now cited as references, giving the authors the opportunity to discuss the fact that Cohn's work was written in the (historical) background of controversial etiology of plague; with alternative proposals that historical outbreaks were due to an infectious agent other than *Y. pestis* and was not a bubonic plague transmitted by rats and fleas despite first microbiological evidence published in 1998 by our team (Drancourt et al ,1998, PNAS) : In “the Black Death transformed” (new reference 40), Cohn presented the hypothesis that so-called second plague pandemic was a different disease than the one during the *Y. pestis* third pandemic including the absence of evidence of any involvements for rats and fleas during the second pandemic outbreaks. In the meantime, second pandemic outbreaks have been undoubtedly confirmed as bubonic plague caused by *Y. pestis* as now confirmed by numerous paleomicrobiological works by several teams in Europe; during which the relative role of rats remains to be documented apart from one rat *Y. pestis* genome (new reference 46).

Our present work is therefore contributive in offering quantitative, controlled data to question alternative scenarios to rats and their ectoparasites; in the absence of any mention of rats (or mice) and their ectoparasites in ancient texts; to be re-interpreted in the context of current knowledge (see Dean and 2018 PNAS, Barbieri et al 2020 CMR, 2020 Lancet Infectious Diseases).

p. 2, plague of 1629-31: note that the most updated study of this plague is Alfani and Bonetti, “A survival analysis of the last great European plagues: The case of Nonantola (Northern Italy) in 1630”, *Population Studies*, 73(1), 2019, pp. 101-118. This article also provides an in-depth analysis of the (probable) means of transmission of the plague in a European historical setting, which is basically along the lines proposed by the authors but which should be taken into account as it is based on yet another kind of external evidence: micro-demographic data from parish records and nominative record linkage.

Authors' answer: The reviewer is perfectly right and this appropriate reference is now cited (as reference 22) and discussed in Lines 64-66 and 324-326.

p. 5, “All these associations suggest that contemporaries considered tatters, clothes, and movable as plague sources that communicated plague across the city”. Well, my point here is

that this was not solely considered to be the cause of plague, but of the vast majority of infectious diseases given the “contagionist” theory which was predominant in the medical science of the time. Hence a finding like this does not tell us anything specific about plague (let alone plague transmission) or at least, it could not immediately and simply be presumed to do so. Again, the authors need to have a good understanding of pre-nineteenth century medical theory in order to truly understand their data & findings, and right now I am quite dubious that they possess such understanding. A recent book which might help (but this is just as a starting point!) is Snowden, *Epidemics and Society: From the Black Death to the Present*, Yale University Press 2019. Note that Snowden is not really updated in his coverage of the plague, hence I am advising the authors to consider him solely as concerns the history of European medical views about epidemics

Authors’ answer: The authors do agree with this remark, they clarify that indeed infection transmission by tatters and clothes, may not have been specific for plague for plague contemporaries; after Snowden's contribution (now listed as new reference 51)

About the fact that the 1629-31 epidemic in north Italy “was never microbiologically confirmed as plague” (p.5) the authors should check Seguin-Orlando, A., et al., "No particular genomic features underpin the dramatic economic consequences of 17th century plague epidemics in Italy", *ISCIENCE*, December 2020. Although I do believe that there are some issues with Seguin-Orlando et al.’s article, it seems to me that it offers some microbiological evidence – which was not really much needed given that based on other material there is little doubt that the 1629-30 epidemic in Italy was caused by “bubonic plague”....

Authors’ answer: The reviewer is perfectly right, this paper has been published along with the editing process of their own manuscript (present manuscript has been submitted on January 10th, 2021; Seguin-Orlando paper has been made available in Google Scholar on April, 23th 2021). Then, obviously the authors now cite (new reference 28) and comment this important contribution which indeed, reinforces present work: indeed, at the time of the present work was undertaken, the authors only predicted that the Milan 1629-1630 site would be a plague site; which is now firmly confirmed by the work of Seguin-Orlando: this is beautifully comforting the usefulness of the method by the authors.

Again at p. 5, maybe I am missing something here, but should the fact that words as “could”, “did”, and “our” are “represented twice”, and “were” even thrice, should tell us something? I can’t see how this is a meaningful finding...

Authors’ answer: Indeed the authors gave crude data as it is the best practice in science: these a priori non-specific words are indisputably “enriched” in plague-related texts over control texts. Then, the authors just did not interpret this observation.

The bottom line of my comments is the following: this article would become stronger, and not weaken, if it accepted the intrinsic limitations of the approach that it proposes. The results are confirmative, but this is not per se a bad thing – given that the methods are novel for the field, having an extra confirmation of what we already thought to know is not simply reassuring: it is useful...

Authors’ answer: The reviewer is just perfectly right and the authors are now clearly stating the potential limitations of present work, emphasizing the contribution of the original method they invented which, hopefully will be used by colleagues in the next future.

Reviewer: 2

Comments to the Author(s)

Rémi and colleagues have presented a quantitative analysis of text-based documents associated with two historic plague outbreaks to search for consistent language that might inform on methods of disease transmission. Texts from the Great Plague of Marseille, 1720 – 1722, are used as “positive controls”, since *Y. pestis* DNA has been identified in human remains associated with the epidemic. A cursory comparison against contemporary texts from non-plague associated episodes was made, though the results of this comparison are discussed only with reference to mention of rats and fleas. The authors then apply this model to texts describing a presumed plague epidemic in Northern Italy (1629 – 1631), which they state has not yet been confirmed to be a plague outbreak via molecular analysis (though I refer the authors to the recently published work by Orlando et al, <https://www.sciencedirect.com/science/article/pii/S2589004221003515>).

Authors' answer: The authors now clarify that the contemporary texts from non-plague associated episodes as stated by the reviewer, were used as negative control texts not only to assess the rat-flea process, but for any variable regarding plague including, at large, the sources and routes of transmission. Also, the paper by Seguin-Orlando paper has been made available in Google Scholar on April, 23th 2021, more than three months after the authors submitted the present paper on January 10th, 2021. Obviously, the authors now cite (new reference 28) and comment this important contribution by Seguin-Orlando and collaborators which reinforces present work: indeed, at the time of the present work was undertaken, the authors only predicted that the Milan 1629-1630 site would be a plague site; which is now firmly confirmed by the work of Seguin-Orlando: this is beautifully comforting the usefulness of the method by the authors.

While the lexicometry approach is both a relevant and welcome addition to methods of plague analysis, the manuscript lacks a clear research statement. Are they attempting to use this model to lend credence to the idea that the plague of Northern Italy was indeed an outbreak of *Y. pestis*? Has that been contested? If so, an exploration of the literature presenting both sides of the debate should be presented. Currently the manuscript reads as though the authors' main goal is to use these methods as an independent line of support for plagues of the second pandemic (those post-Black Death in Europe) having been transmitted via the human louse as opposed to the rat-flea model, the latter of which is well established for plagues of the 20th century. The introduction states that rat-flea models or airborne transmission "may not have been sufficient to fuel the massive medieval mortality, pointing to the complementary role of human lice...". While discrepancies in disease severity between the historic and modern forms of plague are known, their natural extension to human louse transmission is odd and ignores other theories to explain differences in disease progression between the past and today. Differences in susceptibility, both in terms of population health (e.g., frailty, see the work of Sharon DeWitte) and possible genetic changes that are only just beginning to be explored, are deserving of mention, as both would influence the severity of a plague epidemic independent of rat population densities. Also, it should be clear that the majority of the discussants do not dispute the rat-flea model, where the human louse model of plague transmission is a hypothesis discussed by comparatively few.

Authors' answer: As stated above, there is no longer any discussion regarding the Milan site as a plague site, as predicted by our lexicographic analyses and now firmly demonstrated by the paleomicrobiology work by Seguin-Orlando and collaborators.

In fact, a major contribution of the present work, is to propose a completely new method to “read” ancient texts related to plague or not, and this point is now emphasized in the Introduction and Discussion (Lines 70, 339-370) sections of the manuscript.

That said, applying this new lexicographic method to ancient texts reported two different plague sites in Europe, indeed has to provide as neutral as possible data regarding the sources, routes and transmission and overall kinetics of the plague outbreaks in the Modern Times.

That said, I find their approach unconvincing to address their main goal as I understand it. If rats and fleas were not suspected as vectors of infection based on the concepts of contagion understood by the writers, why would they be mentioned in the texts? More discussion is needed to explore the understanding of disease dissemination by the authors of the texts that were queried. I would assume that most doctors understood infectious disease to be carried by air (the miasma theory), where discussion of “stuffs” as vectors of contagion, as they have identified here, bears further contextualisation. Most importantly, the authors essentially argue that rat-flea transmission is a less likely model of plague dissemination since the texts they consulted apparently make no mention of rat mortality (save for one, which they argue is not applicable because the “author spoke of plague in general”... So why is this not valid?). Does that really support the idea that plague was not transmitted by rats? Is absence of evidence truly evidence of absence in this case?

Authors' answer: Once again, the authors now clarify that the major contribution is the method itself, incorporating for the first time to the best of the authors knowledge negative control texts and a comparison between negative controls and examined texts, with clustering analyses as a first round to screen for potential associations which significance are statistically examined versus random.

With respect to data issued from these analyses, authors stuck to data in the perspective of available, factual, previously published data; and cannot endorsed for the two outbreaks here

investigated, the rat-flea model of transmission in the absence of any significant mention of the rats and any rat ectoparasite by the contemporaries; even though the authors do acknowledged the well known limits of interpreting negative results (absence of ..); rat-flea model of transmission which undoubtedly was the one observed in large outbreaks during the third plague pandemics.

Then, the authors clarified that precisely using this new approach, they had exactly no a priori hypothesis regarding the sources and routes of transmission of the plague in these two plague sites.

Authors clearly separated data issued from their analyses in the Results section and illustrating Figures XX, XX and XX; from their interpretation of data in the Discussion section. This interpretation of data has been constrained by the acknowledgements of potential limits of this work, as made remembered by the reviewer (Lines XX); in the strict corpus of available, acknowledged bibliography; and that confrontation leads to give weight to some hypotheses versus others.

Last, the manuscript is over-run with hyperbolic statements that weaken their main arguments. Line numbers did not render properly in my pdf, but here are a few examples:

1. “Unbiased lexicometry” in the title, and the statement, introduction paragraph 3: “we designed a purposefully naïve and unbiased method”. No analytical method is completely absent of bias. In this case biases include observations of the writers, culturally-defined selection of words to describe them, method of translation from French/Italian to English (can true word meaning be obtained through use of modern translation tools for historic texts?), researchers’ choice of word meaning assignment, among many others.

Authors’ answer: The authors do agree with the remark and, accordingly, corrected the title to: “Differential word expression analyses highlight plague dynamics during the second pandemic”, which is purely factual; and acknowledge these limits indicated by the reviewer in the Discussion section (Lines 339-357).

2. Discussion, paragraph 2: “These results resolve a years-long debate as to the etiology of plague in historical sources”. Are the debates really that controversial given the revelations

based on ancient DNA? I fail to see how their method has uniquely resolved anything. If the authors feel differently, the controversies should be explored. Conventional wisdom, including Wikipedia (https://en.wikipedia.org/wiki/1629%E2%80%931631_Italian_plague accessed May, 2021), state that the Italian plague of 1629 was likely caused by *Y. pestis*, transmitted in the bubonic form.

Authors' answer: The reviewer is perfectly right and the authors now removed the sentences related to this somewhat outdated debate.

3. Their central conclusion: they challenge the assumption that “these episodes [plague epidemics] resulted from the reactivation of local plague foci (6) by suggesting human ectoparasites as vectors...”. After reading the manuscript, I see no strong support for the human ectoparasite theory of plague transmission.

Authors' answer: The reviewer is perfectly right, the authors strongly clarified in the revised manuscript, data from the interpretation of the data which, indeed, are not relying at all on any biological investigation.

In short, the approach is relevant, but the conclusions are too strong and lack support.

Presentation of their analysis through transparent acknowledgement of biases and limitations of the method, along with a more balanced presentation of the ideas circulating to explain the differences between historical and modern plagues would make for a more balanced manuscript. A clear and attainable research statement should also be defined.

Authors' answer: The authors acknowledge this positive general comment and took into consideration the above contributive remarks by the reviewer to produce a somewhat more balanced, revised version of the manuscript; even more clarifying what were data and what were interpretations of the data in the background of existing literature.

Reviewer: 3

Comments to the Author(s)

In their submitted manuscript “unbiased lexicometry analyses illuminate plague dynamics during the second pandemic”, Barbieri et al. use automated text mining methods along with statistical analyses on a number of texts describing two historical plague outbreaks as well as ‘negative controls’ to identify unique characteristics of plague texts and to draw conclusions about the epidemiology of historical plague outbreaks of the Second Pandemic in Europe.

The presented approach is unconventional but offers an interesting case study for the possibility of systematic, objective, and automated text mining of digitized historical texts. The analyses are well-documented, and the paper is written in a concise and comprehensible way. However, I have major concerns regarding the study design and the conclusions/interpretations of the results, described in detail below. Therefore, I suggest the send the manuscript back to the authors for major revisions.

Authors’ answer: The authors acknowledge this positive general comment and took into consideration all of the below major remarks by the reviewer, to clarify the manuscript.

Major revisions:

1) My main concern is the poor study design regarding the ‘negative controls’. Currently, the authors use only texts that are unrelated to plague as controls. This is highly problematic, since several of the significant differences in the ‘lexicome’ are likely not associated with plague (*Yersinia pestis*) in particular, but infectious diseases in general. One example is the overrepresentation of “bed” in plague texts which might as well be associated with disease in general. Therefore, it is crucial to include a second negative control with texts about non-plague epidemics, ideally with known causative agent (e.g., historical smallpox or cholera outbreaks). This will add another level of discriminatory power to their analyses, ideally yielding ‘lexicomes’ that are common for infectious diseases and distinct plague ‘lexicomes’.

Authors’ answer: The authors fully agree that choice of the negative controls is crucial, as usually in sciences. In practice, this first ever lexicography study to incorporate “negative control texts” had to build on existing knowledge which, in fact, is very limited in terms of

acute epidemics in France and Italy in the Modern Times. Unfortunately, we cannot include cholera or smallpox because we need control dating from the same period to neutralize words common to the language of the time (. In fact, to the best of our knowledge, only plague has been formally documented in that period of time. Also, the authors took care to incorporate 07 contemporary medical texts as negative controls. At last, these pertinent remarks by the reviewer are now recognized as limits to this pioneering work, in the Discussion section (Lines 339-357).

2) The presented analyses show no overrepresentation of ‘rats’ or ‘fleas’ in plague-related texts, interpreted by the authors as supportive for a human ectoparasite model for plague transmission. Whereas argumenta ex silentio are generally problematic, this case is even worse, as it completely neglects the historical component and is even contradictory. Following the argumentum ex silentio, one could argue as well that the non-overrepresentation of the word ‘bacterium’ is indicative of a non-bacterial causative agent. With this, I just want to emphasize that these Early Modern texts cannot be interpreted without acknowledging the environment in which they were written, at a time when most infectious disease transmission mechanisms were not understood and the miasmatic theory was commonly accepted (see ‘air’, ‘perfume’, ‘miasma’ in the analyses). This interpretation is furthermore contradictory, as (human) ‘lice’ and ‘fleas’ are also not overrepresented. The mere overrepresentation of words like ‘clothes’ or ‘tatters’ are thus not convincing, as they could also emerge from a different, contemporary model of disease transmission related to or equally wrong as the miasmatic theory.

Authors’ answer: The reviewer is mixing two lines of constructive remarks in once.

1. The first remark dealing with the interpretation of negative results, is shared in part by the authors. We do not agree on the extreme example of “bacterium” by the reviewer, this term indeed is not found at all among the texts under evaluation. Concerning the miasmatic theory, this point is now précised at the lines 305 and 353
2. The second remark is dealing with the degree of significance of difference in the expression of words, between controls and plague texts.

3) This leads to the last major concern: the general lack of engagement with medical history. A discussion of the results within the framework of medical history would of

course lower the explanatory power on some aspects, as shown above. On the other hand though, the manuscript would gain not only solidity and soundness by giving treating the historical texts as such, but would also gain an additional perspective on contemporary ideas and theories related to epidemic diseases. With regard to this, also a comparison of ‘medical’ versus ‘lay’ texts on the respective plague outbreaks might be of interest.

Authors’ answer: Indeed, the main objective of the work was to invent a novel method of reading ancient texts, with abstraction of any post-oc cultural reading. That said, the authors do agree with the reviewer that two additional works have to be done: one exposing differences between results here obtained with those that would be obtained with a medical history perspective; and another one using the method here exposed, using another set of ‘lay’ texts.

Minor revisions:

1) In line 94, the authors report that 23.1% of words were excluded due to poor digitization quality. This is tremendous and should not be simply tolerated. I suggest to try re-digitizing the texts, perhaps with different tools, to achieve a better rate.

Authors’ answer: The authors do agree with that remark. Indeed, authors tried pdf-element as another software to digitize texts, yet without any success (four words out of 13 was digitized for example). In the opinion of the authors, Google has the most powerful tools for digitizing ancient texts and unless copying the text manually, no software can do better than Google. We agree that this is a clear-cut limitation that is now acknowledged in the discussion section (Lines 339-357).

2) Several of the texts date decades later than the respective plague outbreak, in one case even around a century. This could be clearly stated in the manuscript, and ideally the analyses should be performed also solely on texts that were not written more than 10 years after the respective outbreak.

Authors' answer: Indeed this information is available in the supplementary dataset S1.

Concerning the plague of Marseille, some texts date from 1820, ie 100 years after the event but it is actually a compilation of texts written in 1720, 1721 and 1722 and published for the 100th anniversary of the plague. The modern introductions to these books as well as the mandatory mention provided by google books have of course been removed for analysis. All the texts used in this study were written by direct eyewitnesses of the plague in Marseille or compiled their testimony without modern commentary (as for the text Plague_FR_15).

Concerning the plague of Milano, a description of all the publication years far from the epidemic is available in the supplementary materials ("Data file S6 (separate file). Historical Italian text related to the 1629-1631 plague epidemic in Northern Italy.).

The date of the writing of the text (when it is known) is now specified in a new column of supplementary dataset S1 text.

3) The manuscript obviously deals with non-English texts written on plague outbreaks in France and Italy. In order to allow historians and philologists to evaluate the results of this paper, I suggest adding a supplementary table presenting all overrepresented words in their original language. Also, words in word clouds should be presented in a searchable text or table format.

Authors' answer: The reviewer is perfectly right, all the requested information (Original words with the traduction, words present in the WordCloud and Nested analysis, Pvalues, Padjust ect...) are available in the Data file S3, and S5 ((Including all the words from plague (Including the Wordcloud and nested words) and control texts in french with the english traduction along with P.value and P.adjust)) concerning the french word and in the Data file S7 and S8 ((Including all the words from plague (Including the Wordcloud and nested words) and control texts in italian with the english traduction along with P.value and P.adjust)).

4) In Table 1, the words overrepresented/nested in plague texts and classified as sources include 'meat' and 'dog' (Marseille) as well as 'feather' (Northern Italy). This should be

discussed in the manuscript, as this might offer interesting insights into contemporary ideas about disease transmission.

Authors' answer: The authors thank the reviewer for further commenting on these data; in Lines 291-301.

5) A recent publication was able to retrieve ancient plague DNA from rat remains associated with an outbreak of the second pandemic (Morozova et al. 2020). This should be at least mentioned in the manuscript, as this at least proves the involvement of rats in historical plague outbreaks.

Authors' answer: The reviewer is perfectly right, this reference is now listed as a new reference 46 and cited in the Discussion section to ascertain that, indeed, rats were possible and probable initiators of the outbreaks, if not major drivers of these outbreaks (Lines 286-288).

6) Also, recent publications of ancient plague genomes associated with the outbreak in Northern Italy around 1630 are not cited: Guellil et al. 2020, San Procolo a Naturno; and Seguin-Orlando et al. 2021, Lariyey-Puy-Saint-Pierre.

Authors' answer: The reviewer is perfectly right and these additional, contributive references are now listed as new references 3 and 28 ; with note that the paper by Seguin-Orlando and collaborators has been made available more than three months after the authors submitted present manuscript for publication.

As the authors answered all the Editor and Reviewers' comments and corrected the manuscript accordingly, they hope that this revised version will be accepted for publication.

Sincerely,

Prof. Michel DRANCOURT, MD, PhD.

Corresponding author

Appendix B

Dear Editor,

Please find enclosed a third, revised version of the manuscript entitled "Differential word expression analyses highlight plague dynamics during the second pandemic." (RSOS-210039.R2) by Rémi BARBIERI and collaborators, along with the answers of the authors to your own comments and the reviewers' comments.

Associate Editor Comments to Author:

Comments to the Author:

In the reviewers' eyes, the paper is much improved and we commend your efforts to address the concerns raised earlier; however, a number of matters remain to be resolved. These appear to be largely linguistic or relating to the authors' choice of phrasing for some parts of the paper (see the reviewer comments for details). We would like you to seek professional language editing support before resubmitting, and making sure you also take into account - and modify as needed - your manuscript to address the remaining issues identified. Please note that the Royal Society has a list of language editing services at <https://royalsociety.org/journals/authors/benefits/language-editing/> - some of which offer discounts to Society authors. Good luck in making these remaining modifications.

Authors' answers. The authors acknowledge the opportunity you are giving them to submit a new version of the manuscript. This version has been corrected by American Journal Experts, one of the two editing professionals among the editing list you kindly provided; so that this revised version hopefully meets the standard of the Royal Society. The correction certificate is enclosed.

Editing Certificate

This document certifies that the manuscript

Differential word expression analyses highlight plague dynamics during the second pandemic.

prepared by the authors

Rémi Barbieri¹, Riccardo Nodari, Michel Signoli, Sara Epis, Didier Raoult, Michel Drancourt

was edited for proper English language, grammar, punctuation, spelling, and overall style by one or more of the highly qualified native English speaking editors at AJE.

This certificate was issued on **October 1, 2021** and may be verified on the AJE website using the verification code **A8EF-3C26-6F9D-7B6D-E12A**.

Neither the research content nor the authors' intentions were altered in any way during the editing process. Documents receiving this certification should be English-ready for publication; however, the author has the ability to accept or reject our suggestions and changes. To verify the final AJE edited version, please visit our verification page at aje.com/certificate. If you have any questions or concerns about this edited document, please contact AJE at support@aje.com.

AJE provides a range of editing, translation, and manuscript services for researchers and publishers around the world.
For more information about our company, services, and partner discounts, please visit aje.com.

Reviewer comments to Author:

Reviewer: 1

Comments to the Author(s)

The current version of this article is a significant improvement over the older one. The article is now more focused on the methodological contribution and avoids making over-bold statements. My main residual concern is with the quality of the text, for two reasons. First, the revision and the general refocusing have left behind some feature of the earlier version – for example, the abstract does not present the article as mostly a methodological contribution, but as a work on plague transmission: as was the case for the earlier version, but not for the current one.

Authors' answers: According to this remark, the authors now amended the abstract by deleting the sentence "while no association was found with rats and fleas, suggesting that during the second plague pandemic, human ectoparasites were

probably major drivers of plague”; indeed, focusing the paper on its methodological contributions (Lines 41).

Secondly, the new parts present many issues with the English. In my revision I had started listing small things to amend, but quickly stopped as the problem is quite systematic. A thorough revision of the English by a native speaker or professional is badly needed before acceptance.

Authors’ answers: The corrections by the reviewer have all been made in the revised version. In addition, this revised version has been corrected by American Journal Experts, one of the two editing professionals among the editing list provided by the Editor. The correction certificate is enclosed.

Apart from issues with the English, I have a list of relatively minor comments:

Line 57, “Readings of second plague pandemics descriptions”, amend to “Readings of second plague pandemic descriptions”

Authors’ answers: Modified accordingly (Line 57).

Line 62, “they may even considered”, amend to “they may even have to be considered” (or at least, this is what I think that the authors mean: there was clearly something missing in the original sentence. Also note that in the light of recent findings from paleo-biology, the authors’ statement in its current formulation might sound too strong)

Authors’ answers: Modified accordingly (Line 62).

Line 63, “could to be related”, amend to “could be related”. IMPORTANT: from this line on I will stop reporting typos and issues with the English – I will instead make the general point that this article still needs to be checked very carefully for issues with the English.

Authors' answers: Modified accordingly (Line 63).

About the plague in Italy in 1629-31 and more generally, about the plague in seventeenth-century Europe, the authors might also check G. Alfani, "Plague in Seventeenth Century Europe and the Decline of Italy: an Epidemiological Hypothesis", *European Review of Economic History*, 2013, 17, 408-430. Also note that sometimes in the current version of the article the 1629-1631 plague is associated to Milan only, while in others it is (correctly) referred to a broader area. Alfani's work mentioned above offers the most updated analysis of the territorial coverage of different plagues in 17th C. Italy. Maybe refer to this plague as "the 1629-31 plague in northern Italy" throughout the article?

Authors' answers: The authors thank the reviewer to rightly point to this contributive reference by G. Alfani (new reference 29). This reference is cited in lines 81 and accordingly the authors took care to correct the inaccurate sentences. The epidemic in Northern Italy is now referred to as "the 1629-31 plague in Northern Italy" as suggested (Lines 191, 249, 383).

p. 11, "texts in the Northern Italy dataset were largely written by nonmedical professionals such as religious figures, historians and economists", first, I struggle with understanding what an "economist" would be in seventeenth-century Italy, and secondly, by checking Dataset S6 I did not find any author that I would describe as an "economist"... Maybe the authors are thinking about something closer to a "philosopher"? (otherwise, I suggest to simply state "professionals such as religious figures or historians").

Authors' answers: Modified accordingly (Table S6)

This being said, as there are some "medical professionals" among the authors of the treatises included in the sample, I think that it would be good to know whether there are differences in the "richness of the repertoire" of Italian professionals and non-professionals working on plague. This kind of test would grant (or not) credibility to the proposed explanation of the difference between Italy and France. If the test fails,

I wonder whether the political (hence, linguistic as well) fragmentation of 17th C Italy, as opposed to 18th C France, offers another viable explanation?

Authors' answers: According to this contributive remark, the authors have analyzed the difference in word richness between the Italian and non-Italian texts; and between texts written by physicians and texts written by non-physicians.

In the new Supplementary Figure 7, featuring a scatterplot 1, each point represents an overrepresented word in the Italian texts; the X axis represents the number of words (in Log2) in the texts written by Italians non-physicians and in Y the number of words (in Log2) in the texts written by Italians physicians. The red dots represent the words in common between the Italian and French texts and the grey dots represent the words found only in the Italian texts. If there was a bias whereby non-physicians would mostly introduce new words that were not or less used by physicians, one would expect that words which are found in common between French (only physicians) and Italian (Physician and non-Physician mix) to end up in the upper left part of the plot (Physician > non-physician). In contrast, red dots are spread along the diagonal line, showing no count bias between physicians and non-physicians text written by Italian authors.

Finally, we assessed whether the words which are more often used by physicians over non-physicians (>2 times) have more chances to be found in common between French and Italian texts, which is not the case ($p_{adj} > 0.05$), suggesting that the differences found between Italian and French texts are unlikely to stem from the distribution of physician vs non-physician witnesses.

Accordingly, the authors corrected the sentence in Lines 235-240, including the contributive hypothesis suggested by the reviewer.

Fisher's Exact Test for Count Data

```
data: res[, Physician - Non_physician] > 1 and res$common_FR_IT
p-value = 0.6248
alternative hypothesis: true odds ratio is not equal to 1
95 percent confidence interval:
 0.4746819 4.2726418
sample estimates:
odds ratio
 1.393343
```

About the role played by animals of various kind (dogs included) in plague transmission, it might be worth checking a recent contribution by Monica Green, which suggests that at the time of the medieval Black Death many animal species might have contributed to spread the disease: Green, "Taking 'Pandemic' Seriously: Making the Black Death Global." In *Pandemic Disease in the Medieval World. Rethinking the Black Death*, Kalamazoo and Bradford: Arc Medieval Press, 2015, pp. 27–61

Authors' answers: The authors thank the reviewer to rightly point to this contributive reference by M.H. Green (new reference 50). This reference is cited in line 296.

p. 15, "1576 Plague", this is best referred to as the "1575-76 plague".

Authors' answers: Corrected accordingly (Lines 326-327).

Reviewer: 3

Comments to the Author(s)

Although I'm not personally convinced regarding the strength of this approach to investigate alternative transmission hypotheses, I acknowledge that the authors toned down their claims, discuss the problems of their approach sufficiently, and backed up their argumentation with references. However, in this context, I cannot follow a change in the abstract which goes rather in the opposite direction. Therefore I request a minor revision in the following sentence:

"Moreover, plague-related words were associated with the words "merchandise", "movable", "tatters", "bed" and "clothes", while no association was found with rats and fleas, suggesting that during the second plague pandemic, human ectoparasites were probably major drivers of plague."

Please change this back to the original "These results support the hypothesis that during the second plague pandemic ..." while maintaining the newly introduced "probably". This reflects much better that the presented study is able to support this pre-existing hypothesis, but cannot suggest this hypothesis on its own.

Authors'answers: The authors took into consideration this remark and the remark by the reviewer 2 and deleted this sentence (Line 39).

Finally, I appreciate that the authors took my previous comments seriously and introduced additional data and discussions which increased the significance of this paper beyond the controversial transmission topic.

Authors'answers: The authors thank the reviewer for their contributive remarks, all incorporated in this revised version, contributing to high standard work and manuscript.

Reviewer: 2

Comments to the Author(s)

I thank the authors for submission of the revised text. In light of their edits I am willing to recommend the manuscript for publication with the exception of two necessary changes:

1. They remove the sentence in the abstract "... while no association was found with rats and fleas, suggesting that during the second plague pandemic human ectoparasites were probably major drivers of plague."

Authors'answers: The authors took into consideration this remark and above remark by the reviewer 3 and deleted this sentence (Line 41).

2. Discussion, fourth last paragraph: "Our results confirmed the features already observed during the second pandemic that plague transmission was probably predominantly human-to-human [15,16,22], probably via human ectoparasites present in clothes"

Authors'answers: Deleted accordingly (Line 334).

The methods employed by the authors do not support these conclusions. As reviewer 1 has suggested, the value of the manuscript is sufficient as a novel methodological approach without the need for conclusions that go beyond the results. That said, a clearer research statement or motivation is still needed, especially since the results are confirmatory of those that have surfaced from other types of data analysis. The authors are of course welcome to describe the human ectoparasite theory as one of the circulating hypotheses on historical plague transmission, but they should not state that the model is supported by their analysis here.

Authors' answers: The reviewer is perfectly right that present paper is a methodological one, and this point has been clarified in the Abstract section and again in the Introduction section (Line 70). Accordingly, the ectoparasite section has been deleted, as an unfocused one.

Greater discussion of medical knowledge of the time and how this might affect the words chosen by the authors of the historical text is still needed. The miasma theory is only obliquely mentioned in the second last discussion paragraph, and should receive greater attention.

Authors' answers: Authors have now discussed these elements into the discussion (lines 302-312)

If Google Books was better than OCR based on tests they performed, the results of these tests should be disclosed.

Authors' answers: According to this remark, authors made an experience (as explained in Lines 390-402) using Plague_FR_02 from the Marseille Corpus, comparing its Google books version and its OCR software PDFelement 6 pro-scanned version. After filtration, the authors retrieved 51,984 words in the Google books version not significantly different from 51,418 words in the PDF element 6 version. Moreover, careful examination of both versions indicated that most of "words" scanned by PDFelement 6 pro were isolated letters present in our dictionary as determinant possessive or verbs. As for an example, Google books yielded one

word "Madame" translated by PDFelement 6 pro into six "words" "M", "A", "D", "A", "M", "E".

This example indicated that using PDFelement 6 pro would imply an extensive control of texts before Deseq2 analysis, at the exact opposite of the method we aimed to develop.

As the authors responded all the Editor and Reviewers 'remarks, and incorporated answers to all these remarks into an English-corrected, revised version, they hope that this revised version will be accepted for publication.

Sincerely,

Prof. Michel DRANCOURT, MD, PhD

Corresponding author

===PREPARING YOUR MANUSCRIPT===

- one version identifying all the changes that have been made (for instance, in coloured highlight, in bold text, or tracked changes);
- a 'clean' version of the new manuscript that incorporates the changes made, but does not highlight them. This version will be used for typesetting if your manuscript is accepted.

===PREPARING YOUR REVISION IN SCHOLARONE===

-- Ensure that your data access statement meets the requirements at <https://royalsociety.org/journals/authors/author-guidelines/#data>. You should ensure that you cite the dataset in your reference list. If you have deposited data etc in the Dryad repository, please include both the 'For publication' link and 'For review' link at this stage.

-- If you have uploaded ESM files, please ensure you follow the guidance at <https://royalsociety.org/journals/authors/author-guidelines/#supplementary-material> to include a suitable title and informative caption. An example of appropriate titling and captioning may be found at https://figshare.com/articles/Table_S2_from_Is_there_a_trade-off_between_peak_performance_and_performance_breadth_across_temperatures_ferrous_aerobic_scope_in_teleost_fishes_/3843624.

Appendix C

Dear Mr Drancourt

On behalf of the Editors, we are pleased to inform you that your Manuscript RSOS-210039.R2 "Differential word expression analyses highlight plague dynamics during the second pandemic." has been accepted for publication in Royal Society Open Science subject to minor revision in accordance with the referees' reports. Please find the referees' comments along with any feedback from the Editors below my signature.

Please submit your revised manuscript and required files (see below) no later than 7 days from today's (ie 18-Nov-2021) date. Note: the ScholarOne system will 'lock' if submission of the revision is attempted 7 or more days after the deadline. If you do not think you will be able to meet this deadline please contact the editorial office immediately.

on behalf of Prof Marta Kwiatkowska (Subject Editor)
openscience@royalsociety.org

Associate Editor Comments to Author:

Please make the final amendments requested by one of the reviewers before resubmitting for final consideration.

Authors' answer: The final amendments were made according to the reviewer's comments, probably allowing forwarding the editorial process.

Reviewer comments to Author:

Reviewer: 3

Comments to the Author(s)

I think the paper improved significantly during the review process, most importantly by focusing on the methodology.

Reviewer: 1

Comments to the Author(s)

I think that the authors did a very good job in making this article the best that it could be. I think that it is now publishable as it is.

Reviewer: 2

Comments to the Author(s)

I thank the authors for submission of their revised manuscript. I have two remaining requests.

1. The authors should amend the statement in the last sentence of their abstract from "... including the sources for the causative *Yersinia pestis*." to something along the lines of "which can inform on the potential sources for the causative *Yersinia pestis*." It is important that the authors apply language to indicate that hypotheses are being developed or further supported by their analytical model, but since their methods offer indirect support with known biases that they now explain, they cannot unequivocally settle outstanding topics of debate such as the source and method of transmission of *Y. pestis* in the second pandemic.

Authors' answer: The reviewer is perfectly, and that sentence has been modified accordingly (Line 42).

2. Line 343 "Rather, the use of these terms suggests a role of for human ectoparasites, including body lice and human fleas, as vectors of *Y. pestis*..." should be changed to: "Rather, the use of these terms is compatible with a role for ectoparasites human ectoparasites, including body lice and human fleas, as vectors of *Y. pestis*...". I remain unconvinced that their analysis here provides any further evidence to support the human ectoparasite theory of plague dissemination, and qualifying language in discussion of this point is needed to make reasonable doubt clear.

Authors' answer: The reviewer is perfectly helpful in accurately refining authors' expression, the sentence has been modified accordingly (Lines 343-345).

The authors would like to thank the reviewers for their constructive comments during the review process which have significantly improved this original work.

Sincerely,

Prof. Michel DRANCOURT, MD, PhD
Corresponding author